# SUFFICIENT CONTEXT: A NEW LENS ON RETRIEVAL AUGMENTED GENERATION SYSTEMS

**Hailey Joren**[*]
UC San Diego
hjoren@ucsd.edu

**Jianyi Zhang**[†]
Duke University
jianyi.zhang@duke.edu

**Chun-Sung Ferng**
Google
csferng@google.com

**Da-Cheng Juan**
Google
dacheng@google.com

**Ankur Taly**
Google
ataly@google.com

**Cyrus Rashtchian**
Google
cyroid@google.com

## ABSTRACT

Augmenting LLMs with context leads to improved performance across many applications. Despite much research on Retrieval Augmented Generation (RAG) systems, an open question is whether errors arise because LLMs fail to utilize the context from retrieval or the context itself is insufficient to answer the query. To shed light on this, we develop a new notion of sufficient context, along with a method to classify instances that have enough information to answer the query. We then use sufficient context to analyze several models and datasets. By stratifying errors based on context sufficiency, we find that larger models with higher baseline performance (Gemini 1.5 Pro, GPT 4o, Claude 3.5) excel at answering queries when the context is sufficient, but often output incorrect answers instead of abstaining when the context is not. On the other hand, smaller models with lower baseline performance (Llama 3.1, Mistral 3, Gemma 2) hallucinate or abstain often, even with sufficient context. We further categorize cases when the context is useful, and improves accuracy, even though it does not fully answer the query and the model errs without the context. Building on our findings, we explore ways to reduce hallucinations in RAG systems, including a new selective generation method that leverages sufficient context information for guided abstention. Our method improves the fraction of correct answers among times where the model responds by 2–10% for Gemini, GPT, and Gemma. Code for our selective generation method and the prompts used in our autorater analysis are available on our github.

## 1 INTRODUCTION

Providing Large Language Models (LLMs) with additional context, such as in Retrieval Augmented Generation (RAG) systems, has led to major improvements in LLM factuality and verifiability when adapting to new domains (Lewis et al., 2020). In the case of open-domain question answering, a retrieval model provides context at inference time in the form of snippets or long-form text (Zhu et al., 2021). Then, the model synthesizes the query along with this added context to generate the answer. Unfortunately, current RAG-based LLMs exhibit many undesirable traits, such as confidently predicting the incorrect answer with retrieved evidence (Mishra et al., 2024; Niu et al., 2024; Ru et al., 2024), being distracted by unrelated information (Cuconasu et al., 2024; Yoran et al., 2024), and failing to properly extract answers from long text snippets (Hsieh et al., 2024; Liu et al., 2024).

The ideal outcome is for the LLM to output the correct answer if the provided context contains enough information to answer the question when combined with the model's parametric knowledge. Otherwise, the model should abstain from answering and/or ask for more information. One core challenge in achieving this ideal outcome is building models that can use the provided context only when it helps answer the question correctly. Several works have investigated this issue by evaluating

---

[*]Work done during an internship at Google.
[†]Work done during an internship at Google.

models in the presence of irrelevant information in the context (discussed in Section 2). However, "relevant information" can range from directly containing the answer to simply being topically related to the question. Even "golden" or oracle documents in datasets vary in how much information they provide about the query, and whether they directly inform the ground truth answer or not. In other words, while the goal seems to be to understand how LLMs behave when they do or do not have sufficient information to answer the query, prior work fails to address this head-on.

As our first contribution, we put forth a new notion of sufficient context. We divide instances into two categories based on whether the context provides enough information to construct an answer to the query. The sufficient context designation is a function of an input pair consisting of one question and the associated context. Crucially, it does not require a ground truth answer. Figure 1 shows examples and a breakdown of model responses after splitting the data based on sufficient vs. insufficient context. To divide the dataset, we use an LLM-based autorater to classify instances as sufficient or not. Here, an *autorater* is a model that evaluates instances based on a property, e.g., a sufficient context autorater.

Using our sufficient context autorater, we uncover new insights into LLM behavior and into existing benchmark datasets. First, we find models generate incorrect answers on a non-trivial fraction of instances that have sufficient context to answer the query. In other words, open-book QA cannot be solved by improving retrieval alone. Second, when given instances without sufficient context, models tend to hallucinate more than they abstain, especially for multi-hop questions. This finding complements prior work, which shows that LLMs are not robust to noisy retrieval (Yoran et al., 2024; Wu et al., 2024). Third, models generate correct answers in many cases, even when the provided context is insufficient. Surprisingly, this remains true after we filter out questions that the model answers correctly in a closed book (w/o RAG) setting. Together, our analysis deepens our understanding of RAG systems by revealing nuances in how models generate responses with retrieval.

As a final contribution, we explore ways to use sufficient context labels to reduce model hallucinations. We implement a new selective generation framework that improves accuracy. We use a smaller, intervention model to determine when the model generates or abstains, providing a controllable trade-off. Moreover, we can combine our method with any LLM, including proprietary models like Gemini and GPT. Our main result is that using sufficient context as an additional signal leads to much higher accuracy over the fraction of answered queries, for most coverage levels and across multiple models/datasets. We also find that fine-tuning open-source models with sufficient context information does not easily reduce the hallucination rate. Instead, for LLama 3.1 and Mistral 3, fine-tuning can lead to a higher abstention rate at the cost of fewer correct answers. Code for our selective generation method and the prompts used in our autorater analysis are available on our github.

To summarize, our main contributions are

1. We define the notion of sufficient context, unifying existing work on relevance for RAG systems. Then, we design a sufficient context autorater (achieving 93% accuracy), enabling us to label instances scalably and to analyze model responses with or without sufficient context.

2. Our analysis leads to several new findings about retrieval-augmented model performance. One takeaway is that SOTA LLMs output correct responses 35–62% of the time with insufficient context. Hence, intervention strategies to increase accuracy should not solely rely on sufficiency.

3. Building on our findings above, we develop an efficient and general method for selective generation, using both confidence and sufficient context signals. Our method improves the fraction of correct answers (among total model responses) by up to 2–10% for Gemini, GPT, and Gemma.

## 2 RELATED WORK

Many papers have shown that reaping the benefits of RAG (e.g., better factuality) will require a deeper understanding of how LLMs respond to variations in the queries and provided context (Asai et al., 2024; Fan et al., 2024; Ram et al., 2023; Rau et al., 2024). We review two main areas. First, much work has evaluated RAG systems with poor retrieval, uncovering cases where LLMs are led astray by irrelevant context. Another line of study aims to reduce LLM hallucinations in RAG settings.

**(Ir)relevant Context.** Prior studies uncover a lack of robustness to imperfect retrieval. However, these studies vary in terms of how they evaluate retrieval quality, without anchoring to a precise "relevance" definition. Shi et al. (2023a) adds sentences to math questions (based on GSM8K) which

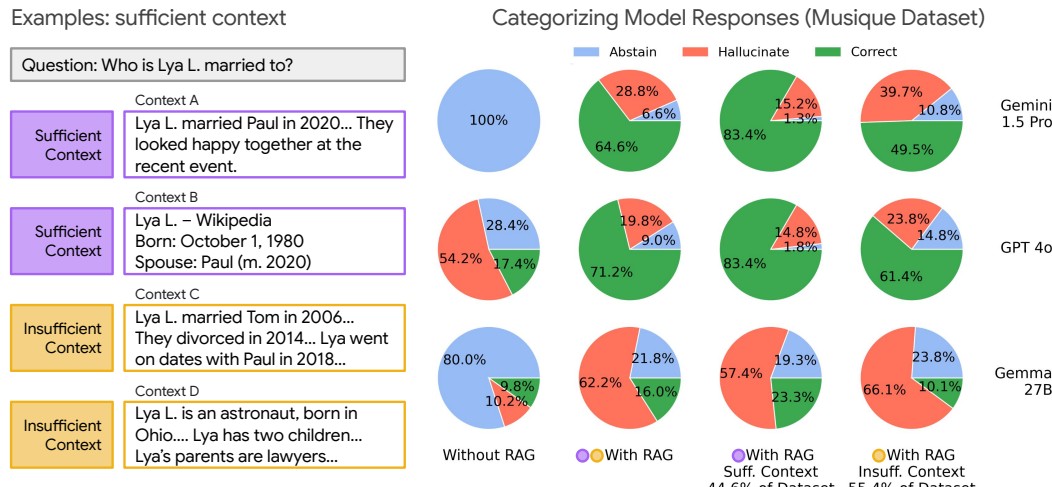

Figure 1: **New insights into RAG systems by looking at whether instances have sufficient context.** On the left, we show examples of sufficient context; on the right, a breakdown of model responses on the Musique dataset. Adding RAG improves the percentage of correct answers. Unfortunately, with RAG, models hallucinate more than abstain, and the insufficiency of the context does not account for this major issue. Also, standard datasets have many instances with insufficient context (here, 55.4%). We include results for other datasets (FreshQA, HotPotQA) in Appendix B, showing similar trends.

should not impact the answer at all (and GSM8K is designed to have sufficient context by definition). Xie et al. (2024) looks at having counter-memory context, by either replacing the entity name with an erroneous one or using an LLM to generate a synthetic context supporting the erroneous entity. Ret-Robust (Yoran et al., 2024) trains a model to be robust to irrelevant context, with an NLI-based entailment to determine relevance, and only uses the relevance scores to influence the training mixture of relevant vs. irrelevant documents. Wu et al. (2024) looks at questions where the LLM gets the answer correct without retrieval and is non-robust to changes in the retrieval. Multiple methods use a model to predict relevance scores (as part of a larger pipeline), without calibration to a formal definition (Wang et al., 2024a; Zhou et al., 2024), including for iterative retrieval (Jiang et al., 2024; Yan et al., 2024). In terms of analysis studies, Cuconasu et al. (2024) distinguishes golden and relevant documents, but simply uses "does not contain the answer" as a proxy for irrelevant context.

**Reducing Hallucinations.** There have also been efforts to improve RAG factuality on open-book QA tasks (Asai et al., 2023; Mineiro, 2024; Simhi et al., 2024; Wang et al., 2024b; Zhang et al., 2024b). The main theme is to improve both the generation and retrieval quality, often by fine-tuning one or more components. Also, since RAG leads to very long contexts, another issue that arises is the "lost in the middle" problem (Hsieh et al., 2024; Liu et al., 2024; Yu et al., 2024). These works start with the premise that the provided query/context should be precisely answerable by the LLM, and hence, only analyze their findings in the sufficient context scenario. Independent of RAG, many papers have studied interventions and tools for calibrating LLM confidence in their responses (Chuang et al., 2024; Kadavath et al., 2022; Yin et al., 2023; Zhang et al., 2024a) and performance across disaggregated subsets of data (Paes et al., 2022; Joren et al., 2023).

## 3  SUFFICIENT CONTEXT

At a high level, our aim is to classify input instances based on whether the context contains enough information to answer the query. We split possible contexts into two cases: (1) **Sufficient Context.** The context is sufficient to answer the query if it contains all the necessary information to provide a definitive answer. (2) **Insufficient Context.** Otherwise, a context is insufficient. A context may also be insufficient if the query requires specialized knowledge that is not provided in the context or if the information in the context is incomplete, inconclusive, or contradictory. In this section, we more thoroughly discuss sufficient context. Then, we show how to accurately and scalably label instances.

## 3.1 DEFINITION OF SUFFICIENT CONTEXT

We first set some notation for a generic open-domain question-answering setting following Trivedi et al. (2020). Consider a dataset $D$ with instances of the form $q = (Q, C; A)$, where $Q$ is the query and $C$ is the context that consists of a set of facts. At inference time, we also consider instances $q' = (Q, C)$ without the ground truth answer, where the goal is to predict an answer $A'$ from $Q, C$, and the model's parametric knowledge. To measure correctness, we compare $A'$ and $A$, where there are many options such as exact match, F1 score, or an LLM-based assessment of answer sameness (we use an LLM). Using this notation, we can now define our notion of sufficient context.

**Definition (Sufficient Context).** An instance $q' = (Q, C)$ has sufficient context if and only if there exists an answer $A'$ such that $A'$ is a plausible answer to the question $Q$ given the information in $C$.

To understand this, we can build on the Attributable to Identified Sources (AIS) framework (Rashkin et al., 2023). Entailment via AIS answers a slightly different question. Namely, given an instance $q = (Q, C; A)$, the entailment objective is to determine the truth value of the proposition: The answer to the question $Q$ is $A$ given the information in $C$. The key difference between entailment and sufficient context is that for sufficient context we do not presuppose that we have the answer $A'$ in advance, only that such an answer exists. Finally, we only consider "plausible" answers, where we mean that $A'$ could be an answer to the question $Q$. For example, if the question asks about a person's birthplace, then $A'$ should be a location. We note that this allows for the possibility that the context contains an *incorrect* answer to the question. This is a key requirement, because (i) we would like to be able to use signal from sufficient context at inference time, where we do not have ground truth answers (see Section 5.1) and (ii) we hope to elucidate findings that are robust to ground truth label noise.

**Remark 1 (Multi-hop queries).** In most benchmark dataset (e.g., Musique, HotPotQA), models are expected to be able to do multi-hop reasoning up to four hops, in which they must combining facts to form the answer. However, they should not infer connections that are not in the context. For example, if "Bob's mother was born in New York" then this does not suffice to say Bob was born in New York. But, if the context also says "Bob's mother is Alice..." and "... all of Alice's children were born in New York" then this instance has sufficient context.

**Remark 2 (Ambiguous queries).** If the query is ambiguous, then the context is sufficient if and only if (i) the context can disambiguate the query and (ii) the context provides an answer to the disambiguated query. For example, the question could be "What sport does Mia play?" and the context could contain both "Mia, from New York, plays basketball..." and "... Mia, from California, plays volleyball." This is sufficient because if the query is referring to either Mia from New York or Bob from California, then the context can answer the question.

**Remark 3 (Ambiguous contexts).** Assume the context contains multiple plausible answers to the query. Then it is sufficient if and only if it also provides enough information to distinguish between queries that would lead to each answer. For example, if the question is "What country does Ali live in?" and the context is "Ali lives in Paris" then this instance does not have sufficient context because it is not clear if Ali lives in Paris, France or Paris, Texas, USA. If the context further contains "This weekend, Ali took the train from Paris to Marseille." Then this becomes sufficient because it is almost certain that Ali lives in France as one cannot take a train from Texas to France.

## 3.2 SUFFICIENT CONTEXT AUTORATER

Next, we consider automating the task of labeling whether instances have sufficient context or not. We investigate two questions: (1) Can today's models achieve high accuracy on a challenging, human-annotated dataset? (2) How does an entailment model compare to general-purpose LLMs? To answer these questions, we evaluate methods on human-labeled data. Table 1 shows that Gemini 1.5 Pro can serve as an accurate autorater to label instances in terms of sufficient context. It achieves 93% accuracy, outperforms other methods, and operates without needing a ground truth answer.

**Sufficient Context Labeled Dataset.** Using the above definition, we construct gold labels for each (`query`, `context`) pair. We did not use ground truth answers or model responses. For the instances, we sample a total of 115 instances (queries, contexts, and answers) from standard benchmarks (PopQA, FreshQA, Natural Questions, EntityQuestions). We design the dataset to be very challenging, including single- and multi-hop questions, as well as adding highly related

Table 1: **Sufficient Context AutoRater.** Evaluating model ability to classify sufficient context on a gold-labeled dataset of 115 `(query, context; answer)` instances. Gemini 1.5 Pro (1-shot) performs the best, while FLAMe can be a cheaper alternative. TRUE-NLI and Contains GT need ground truth (GT) answers, while others only use `(query, context)`. Best in column in bold.

| **Metrics:** 
 **Methods** | F1 Score | Accuracy | Precision | Recall | No GT Answer |
|---|---|---|---|---|---|
| Gemini 1.5 Pro (1-shot) | **0.935** | **0.930** | 0.935 | **0.935** | ✓ |
| Gemini 1.5 Pro (0-shot) | 0.878 | 0.870 | 0.885 | 0.871 | ✓ |
| FLAMe (fine-tune PaLM 24B) | 0.892 | 0.878 | 0.853 | **0.935** | ✓ |
| TRUE-NLI (fine-tune T5 11B) | 0.818 | 0.826 | **0.938** | 0.726 | |
| Contains GT | 0.810 | 0.809 | 0.870 | 0.758 | |

information in the context even if it is not sufficient (e.g., a named entity from the question often appears in the context). We evaluate methods' abilities to classify sufficient context (binary labels).

**Methods: Operating on Query-Context pairs.** We use Gemini 1.5 Pro with either instructions (0-shot) or both instructions and a 1-shot example, held out from our dataset. FLAMe 24B is a general autorater model (Vu et al., 2024), but it has a small context window. For FLAMe, we divide the contexts into 1600 token chunks and ask whether each chunk is sufficient. If any chunk is labeled sufficient, we consider the instance to have sufficient context; otherwise, it's labeled as insufficient. We design prompts (in Appendix C) for both models based on the sufficient context definition above.

**Methods: When a Ground Truth (GT) Answer is Available.** For two baselines (TRUE-NLI, Contains GT), we use answers as an additional input to classify sufficient context. TRUE-NLI is a fine-tuned entailment model (Honovich et al., 2022) that checks if the context entails one of the GT answers. Contains GT checks if a GT answer appears in the context. Comparing to entailment is particularly interesting because if a given answer $A$ is entailed by $(Q, C)$, then the context is also sufficient. On the other hand, the reverse is not true, since the answer $A$ is only one possible choice for $A'$. As one consequence, if the ground truth answer $A$ is incorrect, then it may not be entailed by the context. This happens when a named entity is ambiguous (e.g., two people with the same name), and the GT answer is based on one of the people while the context describes the other.

**Results.** Table 1 shows that Gemini 1.5 Pro (1-shot) performs the best overall in terms of F1 score and accuracy. As expected, TRUE-NLI has higher precision and lower recall: it measures entailment, which implies sufficient context. FLAMe outperforms TRUE-NLI in F1 and accuracy, but lags behind Gemini (1-shot), likely because it is a smaller model. The Contains GT method works surprisingly well, indicating that the presence of a ground truth answer correlates with context sufficiency.

**Discussion.** In real-world scenarios, we cannot expect candidate answers when evaluating model performance. Hence, it is desirable to use a method that works using only the query and context. Among these methods, Gemini 1.5 Pro (1-shot) has high accuracy and balanced precision and recall. Therefore, we use it in Section 4 as our main method for analyzing datasets and model responses. Later, in Section 5.1, we use FLAMe as a computationally efficient autorater to provide a signal for selective generation. Our comparison with TRUE-NLI and Contains GT confirms that classifying sufficient context is a different (and more complex) task than determining entailment.

## 4 A New Lens on RAG Performance

We set out to understand RAG performance by looking at sufficient context. We first analyze datasets (Section 4.1), then we investigate model performance with/without sufficient context (Section 4.2). We qualitatively discuss cases where insufficient context leads to correct model responses (Section 4.3).

### 4.1 Do Benchmark Datasets Have High Sufficient Context?

We introduce the datasets that we use for our analysis. Then, we investigate the percentage of instances in these datasets that have sufficient context (according to our autorater). For our study, we do not aim to optimize the retrieval methods (which could increase the sufficient context percentage).

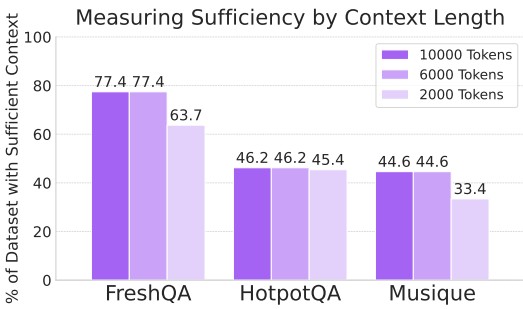

Figure 2: We compare the % of instances that our autorater labels as sufficient across datasets, either with the first 10k, 6k, or 2k tokens of the provided sources. FreshQA has hand-curated URLs that support the answers and exhibits high sufficient context. HotPotQA and Musique have lower sufficient context (and even lower with 2000 tokens). We use 6000 token contexts in the remainder.

This is not the focus of our work, as we wish to understand how models perform with or without sufficient context. Having a mix of both is inevitable in generic RAG systems.

**Datasets.** We consider **FreshQA**, **Musique-Ans**, and **HotpotQA** as a representative spread of open book QA datasets. FreshQA (Vu et al., 2023) evaluates time-sensitive information and has up-to-date URLs that should support an answer to the queries, which we use to construct the `context` (see Appendix A.3 for details on the retrieval). We use the 'True Premise' setting (452 instances), skipping 'False Premise' questions that mislead by design. Musique-Ans (Trivedi et al., 2022) is a multi-hop QA benchmark, created by composing two to four single-hop interconnected questions. Here, 'Ans' is the standard 'answerable' subset. Musique instances have 20 supporting text snippets as sources, which we use as the context. HotpotQA (Yang et al., 2018) is a Wikipedia-based QA dataset, with single- and multi-hop questions. The corpus is a large set of snippets; we retrieve the top 5 as the context (via REPLUG (Shi et al., 2023b) from FlashRAG (Jin et al., 2024)). We randomly sample 500 instances from the development sets of Musique-Ans and HotpotQA for evaluation.

**Sufficient Context % of Datasets.** Figure 2 shows the fraction of instances that our autorater classifies as having sufficient context. We explore three context lengths, ranging from a maximum of 2000 to maximum of 10000 tokens. The motivation behind this is to assess if there is a large change in sufficient context if we were to simply truncate the retrieval (e.g., for models that have small context windows). In general, we see a modest difference from 2000 to 6000 token limit, but effectively none from 6000 to 10000 tokens. FreshQA has the highest sufficient context percentage, which makes sense as the context comes from oracle supporting documents. The lower sufficient context in Musique is perhaps surprising, given that the retrieval is fixed as part of the dataset. From the results in Figure 2, we truncate at 6000 tokens for all three datasets in the remainder of the paper.

### 4.2 INITIAL FINDINGS BASED ON SUFFICIENT CONTEXT

In general, the ideal behavior for a language generation model is to answer questions correctly when possible and to otherwise abstain. RAG seeks to move models towards this desired behavior, such that the provided context shifts hallucinations to correct answers, or to abstentions if needed. We analyze several cases to assess how far we are from this ideal trade-off.

**Experimental Set-up and LLMEval.** We employed a basic chain of thought (CoT) prompting approach, with the prompt structure and further information detailed in Appendix C.4. We then processed the outputted answers to identify matches between the response and any of the ground truth answers. Responses where a clear correct match could not be determined were processed through the LLMEval pipeline using a zero-shot approach, with the prompt based on Krishna et al. (2024) (see Appendix C.3). Then, for each example, we can rate it as "correct" or "abstain" or "hallucinate" depending on the LLMEval output. We use an LLM for evaluation instead of checking for an exact match because it is more robust to syntactic variations. See Appendix B.3 for details and examples.

**Models Abstain Less with RAG.** While overall performance improves with RAG, the introduction of additional context paradoxically reduces the model's ability to abstain from answering when

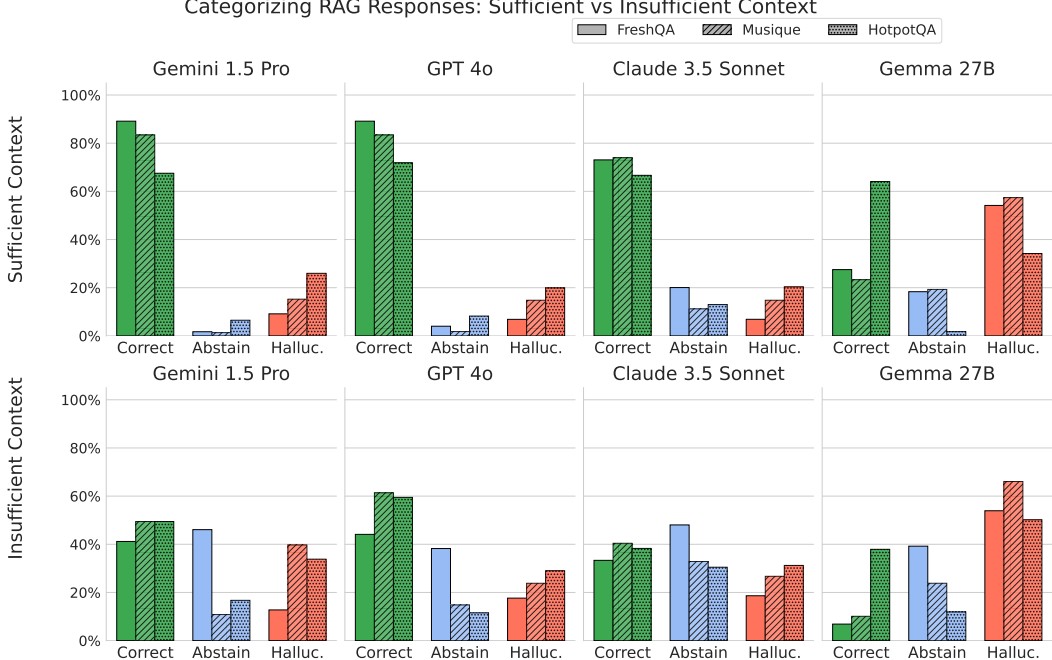

Figure 3: **Model Performance on Datasets Stratified by Sufficient Context.** Given sufficient context, models have a higher correct percentage on these challenging datasets. Performance drops, but the models are still able to answer a large portion of questions correct without sufficient context. One prevailing issue is that all models hallucinate rather than abstain in many cases with insufficient context. The smallest model Gemma 27B struggles to avoid hallucinations given insufficient context.

appropriate. Without RAG, Claude 3.5 Sonnet abstains on 84.1% questions, while with RAG, the fraction of abstentions drops to 52%. Similarly, GPT 4o's abstention fraction moves from 34.4% to 31.2% and Gemini 1.5 Pro's drops from 100% to 18.6%. This phenomenon may arise from the model's increased confidence in the presence of any contextual information, leading to a higher propensity for hallucination rather than abstention.

**Models Hallucinate with Both Sufficient and Insufficient Context.** Considering Figure 3, models generally achieve higher accuracy with sufficient context (higher **green bars**, top row) than without sufficient context (lower **green bars**, bottom row). However, looking at each row separately, we discover several findings. First, in the sufficient context case (top row), we see that models hallucinate more than they abstain (**red bars** are higher than **blue bars**, usually). The trend holds across all three datasets. Moving to insufficient context (bottom row), we find a different distribution of model responses, with more abstentions and hallucinations. This tendency varies notably across different models. For instance, Claude abstains more (higher **blue bars**) with insufficient context, but answers fewer questions correctly (lower **green bars**) than Gemini and GPT. These differences underscore the potential for improvement in both retrieval and reasoning capabilities. Overall, Gemma has much more hallucinations (higher **red bars**) than the other models, except for HotPotQA, where we attribute the higher accuracy to the smaller retrieved contexts.

### 4.3 QUALITATIVELY ANALYZING RESPONSES WITH INSUFFICIENT CONTEXT

One curious observation in our analysis is the ability of models to sometimes provide correct answers even when presented with insufficient context. For example, from Figure 3, all three models are able to correctly answer upwards of 35% of instances with insufficient context on HotpotQA. A natural assumption is that the models already know the answer from pre-training, and they can generate a correct response from parametric memory. However, this only explains part of the story.

Looking deeper, we provide a qualitative categorization in Table 2 of instance types where our autorater labels an instance as insufficient context, while the LLM evaluator marks the model answer

| Instance type | Why model may be correct | Example |
|---|---|---|
| Yes/No question | 50% chance of correct | **Q:** Is there a total eclipse in the United States this year? |
| Limited choice | Some chance of correct | **Q:** Which band has more members, Chvrches or Goodbye Mr. Mackenzie? |
| Multi-hop: fragment | Use parametric inference | **Q:** Who did the original voice for the character whose series Mickey's Safari in Letterland is from? *Context says Mickey's Safari is a video game and Walt Disney voices Mickey Mouse in cartoons. Must infer the game is in the Mickey Mouse series.* |
| Multi-hop: partial | Use parametric knowledge | **Q:** Claudine's Return starred the actress who played which role on "Married...with Children"? *Context lists actresses but not their roles in "Married...with Children". Must know extra facts.* |
| Too many hops | Execute complex reasoning | **Q:** How many cyclists have won all three of women's cycling Grand Tours equivalents in the same year? *Context requires cross-referencing lists of events and lists of winners while tracking winners by year.* |
| Ambiguous query | Guess right interpretation | **Q:** Who is the spouse of a cast member from King of the Mountain? *Context has many cast members and query/context do not specify which spouse to answer about.* |
| Rater error | Mislabel insuff. or correct | — |
| Closed-book correct | Known from pre-training | — |

Table 2: **Qualitative Analysis of Correct Answer & Insufficient Context.** Examining model responses across datasets, we identify common cases where the model generates a correct answer even though our autorater labels the instance as insufficient. We categorize such instances into eight types, as well as provide examples. Given that models are also correct on many questions in the closed-book setting, we believe this mostly explains the 35–62% correct rate with insufficient context.

as correct. For example, one type accounts for when the provided context is not sufficient to answer the query, but it bridges gaps in the model's knowledge. Another type is when the retrieved information clarifies ambiguities inherent in the question (without answering the question). Finally, we also have the times where either the autorater or the evaluator model makes an error. We note that our analysis expands on prior work by Yoran et al. (2024), who also find a large fraction of cases where the model is correct with RAG (but not without) even though the context does not contain the answer.

We additionally investigated cases where the autorater labels an instance as having sufficient context while the LLM evaluator marks the answer as incorrect. One source of these discrepancies occurs when the ground truth answer conflicts with the answer provided in the source. This represents a key difference from methods that measure entailment, where context is evaluated relative to a specific ground truth answer (see Table 1). Another source of errors arises when the autorater correctly identifies that the necessary information is present, but the model fails to properly compose the information (e.g., in multihop questions or questions requiring arithmetic). In a substantial number of cases, however, determining the source of the error proves challenging.

## 5 Techniques to Reduce Hallucinations with RAG

From our previous analysis, we have seen that models may hallucinate rather than abstain and that this happens more with RAG than in a closed-book setting. A natural next question is whether we can prompt or fine-tune a model to perform closer to the ideal case. Can we steer the model to either output the correct answer or abstain, while hallucinating an incorrect answer as little as possible?

### 5.1 Selective RAG Using Sufficient Context Signal

One simple solution to improving RAG performance would be to use the sufficient context autorater to abstain given insufficient context. However, this heavy-handed approach can lower overall performance, since all models answer some questions correctly even with insufficient context, as described in Table 2 and demonstrated in Figure 3. Instead, we propose a method for combining

the sufficient context autorater outputs with model self-rated confidence scores to tune a selective accuracy-coverage trade-off, where "coverage" denotes the portion of inputs on which the model does not abstain. Specifically, we use these signals to train a simple linear model to predict hallucinations, and then use it to set coverage-accuracy trade-off thresholds.

This mechanism differs from other strategies for improving abstention in two key ways. First, because it operates independently from generation, it mitigates unintended downstream effects, whereas strategies like fine-tuning to improve abstention can inadvertently worsen performance on certain inputs (see Section 5.2). Second, it offers a *controllable* mechanism for tuning abstention, which allows for different operating settings in differing applications, such as strict accuracy compliance in medical domains or maximal coverage on creative generation tasks.

**Abstention Signals** We utilize two main signals for abstention: the self-rated probabilities as in Li et al. (2024); Kadavath et al. (2022) and the sufficient context autorater. For the self-rated probabilities, we use two strategies: P(True) and P(Correct). P(True) requires sampling answers from the model multiple times, and then prompting the model multiple times to label each model as correct or incorrect, resulting in a final probability of correctness associated with each question as in Kadavath et al. (2022). For proprietary models, where extensive querying is prohibitively expensive, we use P(Correct) instead. We adapt the probability-generating prompt from Li et al. (2024) to obtain the model's response and its estimated probability of correctness. For the sufficient context signal, we use the binary label from an autorater. Our hypothesis is that combining these signals should lead to more effective abstention, particularly in cases where the context is insufficient.

**Methods.** We calculate P(True) by sampling 20 responses for each question and querying the model 5 times to evaluate whether the answer is correct or incorrect (without using the ground truth) as in Kadavath et al. (2022). For P(Correct), the prompt requests the most likely and second most likely answers along with their probabilities. We use string matching to extract the response and self-predicted probability, keeping the one with the highest probability. To determine sufficient context, we use FLAMe, a small and efficient model for determining the sufficient context label. We divide the retrievals into chunks of 1600 tokens to fit in the context window and label the context as sufficient if any of these chunks are sufficient.

We combine the binary sufficient context label with the self-rated answer probability (P(True) for open-source models or P(Correct) for proprietary models) in a simple logistic regression model to predict hallucinations with 100 iterations of random hyperparameter search. At inference time, we use the logistic regression model scores to threshold the outputs, abstaining when the score is below a chosen threshold as in Joren et al. (2024). We measure the added value for selective accuracy of the sufficient context signal (purple line in Figure 4) by comparing it with the model self-rated confidence alone (gray line).

**Results.** We find that our approach leads to a better selective accuracy-coverage trade-off compared to using model confidence alone. In particular, see gains of over 10% for Gemma 27B on HotpotQA in the highest accuracy regions, and gains of over 5% for Gemini 1.5 Pro on the same dataset near the 70% coverage region. These gains are less pronounced on datasets with lower overall accuracy, such as when using Gemma 27B on Musique. In this scenario, the low overall performance (18.4%) likely means that most of the predictive gains are seen by using the self-rated confidence to predict errors for the majority of samples. As a result, there is no added benefit from the sufficient context signal.

**Discussion.** As expected, we see a downward trend in which higher coverage leads to lower selective accuracy for both methods. We conclude that the selective generation mechanism with sufficient context has an added benefit for accuracy-coverage trade-offs compared to self-rated confidence alone. As a prerequisite for our method, models should have a non-trivial accuracy on sufficient and insufficient context instances. Then, intuitively, we can prioritize answering questions the model is likely to get right before those the model struggles with. While ordering examples is impossible in real settings, we can estimate a coverage level and use a threshold to choose when to answer.

## 5.2 FINE-TUNING

We also consider fine-tuning models to increase their ability to abstain instead of outputting an incorrect answer. To do so, we train the models with some examples that contain "I don't know" instead of their original ground truth answer. The intuition here is that training explicitly on such

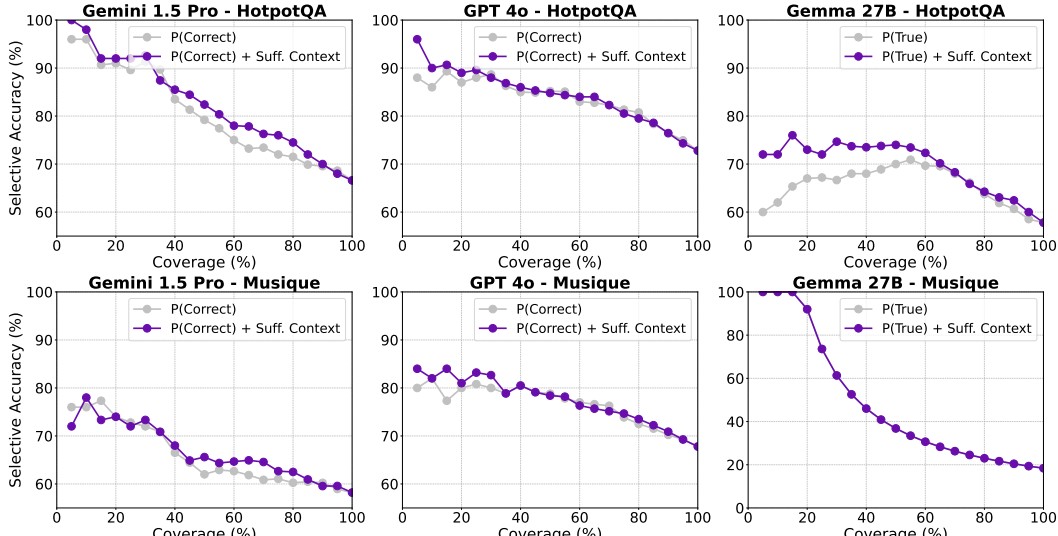

Figure 4: **Selective Generation: Coverage vs. Selective Accuracy.** For selective generation, we use a linear combination of sufficient context and self-rated confidence (**purple**) or confidence alone (gray). The x-axis shows coverage (% of questions answered); the y-axis shows accuracy at each coverage (# correct / # answered). The combined approach matches or outperforms the baseline confidence-only method, especially on HotpotQA, where our method improves accuracy for most coverages. For Gemma 27B on Musique, the methods are identical (coeff. for stuff. context is 0).

inputs could encourage the model to abstain instead of hallucinating. We also consider multiple models (Llama 3.1 8B and Mixtral 3 7B) and multiple settings, such as only changing the answers when the example has insufficient context. We present full details in Appendix B.1. The main takeaways are that fine-tuned models (i) have a higher rate of correct answers in many cases, but (ii) still hallucinate quite often and more than they abstain. Overall, it is likely possible to use fine-tuning to steer the model towards better abstention and correctness, but more work is needed to determine develop a reliable strategy that can balance these objectives.

## 6 CONCLUSION

Our work provided a new lens on LLM responses in RAG systems centered around our notion of sufficient context. We constructed a sufficient context autorater, which enabled scalable insights into model performance on different types of instances. Our analysis revealed that even with sufficient context, LLMs frequently hallucinate answers. We also found, surprisingly, many cases where a model will output a correct answer with access to only insufficient context. Qualitatively, we categorized such instances, leading to a fuller picture of ways context can be useful. Finally, we demonstrated a general-purpose selective generation method, which applies to Gemini, GPT, and Gemma, and can reduce hallucinations by 2–10% on queries that the model answers.

**Limitations.** Our analysis focuses on QA datasets, but summarization tasks also utilize context, which may or may not be sufficient. For example, models may behave differently on the prompt "Summarize the reviews of 5-star hotels in Mallorca" depending on whether the context mentions the hotel reviews, whether they are for 5-star hotels, etc. Another shortcoming is an exploration of how often different retrieval methods lead to sufficient context. Also to achieve the best performance, we could have used our autorater to iteratively judge whether to retrieve more or answer the question.

**Future Work.** One direction is a fine-grained sufficient context autorater, which outputs a score instead of a binary label. This could be useful for ranking contexts after the retrieval step. Another direction is to extend the definition of sufficient context to multi-modal RAG settings, such as for visual QA (images) or document QA (pdf files). Finally, our selective generation results suggest that there is room for improvement in reducing hallucinations by using auxiliary signals from the inputs.

ACKNOWLEDGMENTS

We thank Hrishikesh Garud, Vikram Gopali, Xun Sun, and Bruce Wang for annotating data. We thank Ranjay Krishna and Jacob Eisenstein for helpful discussions. We also thank Alyshia Olsen for help with the figure design and color palette. We thank the anonymous reviewers for suggestions to improve the presentation.

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

# A SUPPORTING EXPERIMENT INFORMATION

We provide sufficient details to reproduce all of our experiments. We list all of the prompts that we use in the next section.

## A.1 MODELS

**GPT 4o.** We use the API-accessible `gpt-4o-2024-08-06` model, released on August 6, 2024 (Achiam et al., 2023).

**Gemini 1.5 Pro.** We use the API-accessible `gemini-1.5-pro-0514` model, released on May 14, 2024 (Gemini Team et al., 2023).

**Claude 3.5 Sonnet.** We use the only API-accessible `claude-3-5-sonnet-20240620` model, released on June 20, 2024 (Anthropic, 2024).

**Gemma 2 27B.** We use the publicly available instruction tuned `gemma-2-27b-it` model, released on Jun 27, 2024 (Gemma Team et al., 2024).

**Llama 3.1 8B.** We use the publicly available instruction tuned `Llama-3.1-8B-Instruct` model, released on July 23, 2024 (Dubey et al., 2024).

**Mistral 3 7B.** We use the publicly available instruction tuned `Mistral-7B-Instruct-v0.3` model, released on May 22, 2024 (Jiang et al., 2023).

**FLAMe.** We use the published `FLAMe-RM-24B` model (Vu et al., 2024).

**TRUE-NLI model.** Calculated the maximum probability over the chunks in the context. Use a threshold of 0.05, where if the maximum probability is higher then this, then we classify as 'sufficient context'. The threshold of 0.05 achieved the highest F1 score on our human labeled dataset. We use the `t5_xxl_true_nli_mixture` version of their model (Honovich et al., 2022).

## A.2 FINE-TUNING SETTINGS

In our fine-tuning setup, we employed the LoRA adaptation technique (Hu et al., 2022) to fine-tune two models: LLaMA 3.1 8B instruct and Mistral-7B-Instruct-v0.3[1]. We used either a 2,000-example random subset sampled from the training set of the Musique-Ans dataset or from the development set of the HotPotQA data[2]. The prompt template for finetuning is provided in Appendix D.

For the LoRA parameters, we set the rank to 4 and alpha to 8 for all experiments. The models were fine-tuned over 2 epochs with a batch size of 16 and a learning rate of $1 \times 10^{-5}$. We note that the training was not smooth, and different checkpoints led to very different results. To be systematic, we chose the best checkpoint in terms of Correct % after either 1 or 2 epochs (where for Musique it turned out to be after 1 epoch, and for HotPotQA we found that 2 epochs was better).

## A.3 DATASETS

We sample 500 examples from HotPotQA and Musique-Ans dev sets, following prior work. We use all 'True Premise' questions from FreshQA.

**Retrieval for HotpotQA.** We adopt the FlashRAG framework (Jin et al., 2024) to implement our Retrieval-Augmented Generation (RAG) process. Our retrieval corpus is based on the wiki-18 dataset, utilizing 'intfloat/e5-base-v2' from Hugging Face's model hub as a Dense Retriever [3]. For each query, we retrieved the top 5 documents, which are subsequently concatenated with the query and placed within a prompt template for inference.

To explore advanced retrieval techniques, we also evaluated the REPLUG (Shi et al., 2023b) method. REPLUG enhances the generation quality by prepending each retrieved document individually to the

---

[1]Available at `huggingface.co/mistralai/Mistral-7B-Instruct-v0.3`

[2]We use dev set for HotPotQA since the training set had a much different distribution of sufficient context examples. Namely, we found the train set to be over 88% sufficient context, while the dev set was only 44%.

[3]`huggingface.co/intfloat/e5-base-v2`

input context and ensembling output probabilities across different passes. The REPLUG method is also implemented based on the FlashRAG framework (Jin et al., 2024).

**Retrieval for FreshQA** We use the urls provided in the FreshQA dataset as retrieval for the `context`. We scraped each url and discarded extra HTML content such as headers, footers, and navigation. We include the title of each webpage in the text, convert any included tables to markdown, and include the table title immediately before the table in the text. When splitting the tables and text for smaller context windows, we keep tables and sentences intact when possible. For large tables that require splitting, we duplicate the table row column headers to include them in each chunk.

Table 3: **Fine-tuned (FT) Llama 3.1 8B Instruct and Mistral 3 7B Instruct models**. We compare closed book and vanilla RAG with three FT settings, measuring % Correct (%C), % Abstain (%A), and % Hallucinate (%H). Also, "idk" means we change the answer in training samples to be "I don't know" instead of the given answer (either for 20% of random examples, or 20% of examples with insufficient context). Best %C for each model/dataset in bold.

| Model | Variant | RAG | Musique | | | HotPotQA | | |
|---|---|---|---|---|---|---|---|---|
| | | | %C | %A | %H | %C | %A | %H |
| Llama | Closed Book | | 2.8 | 76.4 | 20.8 | 18.8 | 57 | 24.2 |
| " | Vanilla RAG | ✓ | 19.6 | 53.6 | 26.8 | 36.8 | 40.4 | 22.8 |
| " | FT GT answer (Data Mix 1) | ✓ | **29.2** | 31.4 | 39.4 | **39.4** | 27.6 | 33 |
| " | FT idk 20% rand. (Data Mix 2) | ✓ | 26.8 | 37.2 | 36 | 39.2 | 28.6 | 32.2 |
| " | FT idk 20% insuff. (Data Mix 3) | ✓ | 25 | 38.8 | 36.2 | 38 | 30.4 | 31.6 |
| Mistral | Closed Book | | 6.6 | 29.8 | 63.6 | 32 | 7.6 | 60.4 |
| " | Vanilla RAG | ✓ | 28.8 | 11.8 | 59.4 | **46.6** | 9.2 | 44.2 |
| " | FT GT answer (Data Mix 1) | ✓ | **31.4** | 0 | 68.6 | 43.4 | 0 | 56.6 |
| " | FT idk 20% rand. (Data Mix 2) | ✓ | 23 | 1.2 | 75.8 | 41.6 | 0.8 | 57.6 |
| " | FT idk 20% insuff. (Data Mix 3) | ✓ | 23 | 2.2 | 74.8 | 41.2 | 2 | 56.8 |

## B    ADDITIONAL RESULTS

### B.1    FINE-TUNING FULL RESULTS

One aspect of our selective generation framework is that we use FLAMe, a 24B model, to provide sufficient context labels. However, we would incur significant overhead if we used a 24B model to improve the generation of a much smaller LLM. Instead, we try directly fine-tuning Llama 3.1 8B and Mistral 3 7B to increase accuracy with retrieval. Specifically, we experiment with different data mixtures to encourage the model to output "I don't know" instead of generating an incorrect response.

**Fine-tuning Data.** We repeat the following process separately for each of the Musique-Ans and HotPotQA datasets to create three mixtures of training data with different answers for each dataset. First, we sample 2000 instances. Then, for Data Mix 1, we fine-tune on these instances and keep their given ground truth answer. For Data Mix 2, we choose 400 examples (20%) at random and change the answer to "I don't know" before fine-tuning. For Data Mix 3, we instead randomly choose 400 examples (20%) that our autorater labels as insufficient context and change their answer to "I don't know" while keeping the other answers as the ground truth. Our hypothesis is that fine-tuning on Data Mix 2 and 3 should steer the model to abstain more and hallucinate less than with Data Mix 1.

**Models, Methods, Metrics.** Using the three data mixtures described above, we fine-tune the Llama 3.1 8B Instruct and Mistral 3 7B Instruct models model with LoRA (details in Appendix A.2). At inference time, we use the standard RAG setup where we add context to the prompt. As baselines, we also evaluate the model without fine-tuning in both the closed-book setting (w/o RAG) and the open-book setting (Vanilla RAG). Consistent with the prior experiments, we use an LLM with ground truth answers to classify responses as Correct (%C), Abstention (%A), or Hallucination (%H).

**Fine-tuning Results and Discussion.** Table 3 shows our experimental results. We verify that the FT variants have a higher rate of generating correct answers (%C) compared to closed-book and Vanilla RAG in three of 4 cases. On the other hand, refuting our hypothesis, Data Mix 2 and 3 do not lead to more abstentions than Vanilla RAG. But, they do abstain more than with Data Mix 1, showing the impact of adding "I don't know" in the training set. In general, FT models using RAG output incorrect answers (%H) at least 31% of the time, and often more than they abstain (%A).

### B.2    PERFORMANCE BREAKDOWN BY SUFFICIENT CONTEXT

We explore RAG performance by different models for various RAG benchmark datasets. Here, the first column shows performance without RAG (closed-book) while the second column shows performance with RAG (open-book). To better understand RAG performance, we use our sufficient context autorater to stratify the retrieval augmented generation (RAG) datasets into sufficient and

insufficient context. The third and fourth columns show the performance of the second column stratified by sufficient vs insufficient context respectively.

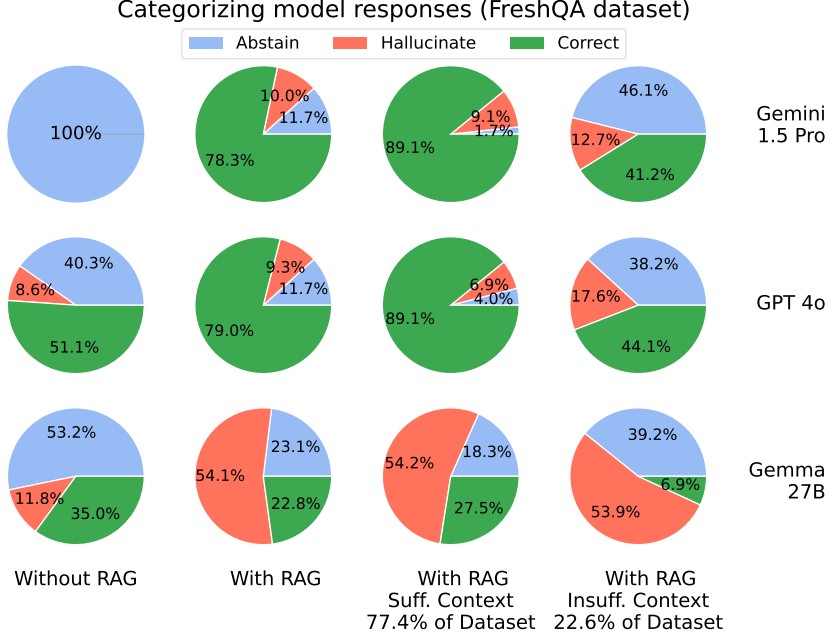

Figure 5: Correct, hallucination, and abstention fractions across models for dataset FreshQA, stratified by sufficient context. FreshQA includes hand-curated source URLs, which explains the larger percentage of sufficient context (77.4%). FreshQA also specifically explores questions with answers that change based on the question's timestamp, which may explain the frequent abstentions without RAG (100% for Gemini 1.5 Pro).

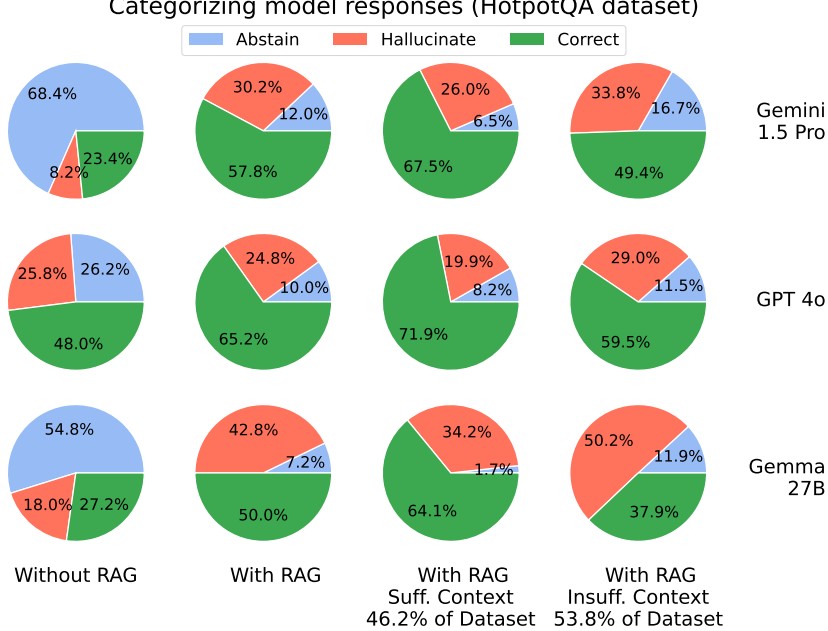

Figure 6: Correct, hallucination, and abstention fractions across models for dataset HotpotQA, stratified by sufficient context. HotpotQA includes questions that are more likely to be answerable without context (e.g., yes or no questions, multiple choice questions, or questions with answers that might be answerable due to pre-training). This explains the higher fraction of correct answers without RAG (e.g., 48.0% for GPT 4o).

Table 4: **Performance Analysis of RAG Systems Using Human-Annotated Sufficient Context Labels.** These tables include results on a curated set of challenging context-dependent questions. Table (a) shows that while larger models generally achieve higher accuracy with sufficient context (present in 54.8% of cases), even top performers exhibit a 14-16% error rate. Table (b) reveals that with insufficient context (45.2% of cases), models predominantly abstain from answering (50-73% of instances), though significant hallucination rates (15-40%) persist. These patterns of context-dependent performance and hallucination risk are consistent with our analyses of HotpotQA, FreshQA, and Musique datasets, despite variations in absolute performance due to different task complexities.

(a) Performance with Sufficient Context (54.8% of Dataset)

| Model | % Correct | % Abstain | % Hallucinate |
|---|---|---|---|
| Gemini 1.5 Pro | 84.1 | 1.6 | 14.3 |
| GPT 4o | 82.5 | 4.8 | 12.7 |
| Claude 3.5 Sonnet | 85.7 | 11.1 | 3.2 |
| Gemini 1.5 Flash | 77.8 | 4.8 | 17.5 |
| Gemma 27B | 71.4 | 3.2 | 25.4 |

(b) Performance with Insufficient Context (45.2% of Dataset)

| Model | % Correct | % Abstain | % Hallucinate |
|---|---|---|---|
| Gemini 1.5 Pro | 9.6 | 50.0 | 40.4 |
| GPT 4o | 23.1 | 61.5 | 15.4 |
| Claude 3.5 Sonnet | 9.6 | 53.8 | 36.5 |
| Gemini 1.5 Flash | 7.7 | 73.1 | 19.2 |
| Gemma 27B | 9.6 | 55.8 | 34.6 |

### B.3 COMPARISON OF QA EVALUATION METRICS

We compare two the LLM-based QA Evaluator (LLMEval) used in the paper with a deterministic lexical matching metric (Contains Answer). The Contains Answer metric labels responses based on whether they contain the exact ground truth answer, while LLMEval uses an LLM to assess semantic correctness.

Table 5 presents model performance across three datasets (FreshQA, Musique, HotpotQA), split by our sufficient context autorater. The results show Contains Answer is generally stricter than LLMEval, though both metrics reveal similar patterns in model behavior.

Table 5: **Comparison of evaluation metrics across models and datasets.** We show results for checking whether the response contains one of the ground truth answer strings ("Contains"), where we report the % of responses that contain an answer. We compare this to our LLMEval method that uses an LLM to evaluate if the response is correct, abstain, or hallucinated, and we report % correct.

| | | FreshQA | | Musique | | HotpotQA | |
|---|---|---|---|---|---|---|---|
| Model | Context | Contains | LLMEval | Contains | LLMEval | Contains | LLMEval |
| Gemini 1.5 Pro | Suff | 80.3% | 89.1% | 60.1% | 83.4% | 47.6% | 67.5% |
| | Insuff | 31.4% | 41.2% | 33.6% | 49.5% | 34.2% | 49.4% |
| GPT-4 | Suff | 84.3% | 89.1% | 64.6% | 83.4% | 52.4% | 71.9% |
| | Insuff | 36.3% | 44.1% | 44.4% | 61.4% | 46.1% | 59.5% |
| Gemma 27B | Suff | 26.9% | 27.5% | 10.8% | 23.3% | 40.7% | 64.1% |
| | Insuff | 11.8% | 6.9% | 7.2% | 10.1% | 22.7% | 37.9% |
| Claude 3.5 | Suff | 67.9% | 73.1% | 48.9% | 74.0% | 46.3% | 66.7% |
| Sonnet | Insuff | 26.5% | 33.3% | 19.9% | 40.4% | 29.0% | 38.3% |

The Contains Answer metric exhibits several characteristics when compared to LLMEval:

1. Different formatting affects matching:

```
Q: What date did the creator of Autumn Leaves die?
Ground Truth: 13 August 1896
Response: August 13, 1896.
Contains Answer: False
LLMEval: Correct
```

2. Semantic equivalents are not captured:

```
Q: What former Los Angeles Lakers majority owner is the
father of Jeanie Marie Buss?
Ground Truth: Gerald Hatten Buss
Response: Jerry Buss.
Contains Answer: False
LLMEval: Correct
```

3. Partial matches can be marked as correct:

```
Q: What is Amazon Prime Video's most watched premiere ever?
Ground Truth: The Rings of Power
Response: The series explores the forging of the Rings of Power,
the rise of Sauron...
Contains Answer: True
LLMEval: Hallucinate
```

The LLM QA evaluator provides several practical advantages:

- Handles variations in model verbosity and formatting
- Distinguishes between correct, abstain, and incorrect responses
- Enables efficient evaluation across multiple datasets

Our analysis shows two key findings that are consistent across both metrics: LLMs (i) exhibit hallucination even with sufficient context and (ii) struggle to abstain with insufficient context.

## C PROMPTS

### C.1 SUFFICIENT CONTEXT AUTORATER PROMPT

You are an expert LLM evaluator that excels at evaluating a QUESTION and REFERENCES. Consider the following criteria:
Sufficient Context: 1 IF the CONTEXT is sufficient to infer the answer to the question and 0 IF the CONTEXT cannot be used to infer the answer to the question
Assume the queries have timestamp <TIMESTAMP>.
First, output a list of step-by-step questions that would be used to arrive at a label for the criteria. Make sure to include questions about assumptions implicit in the QUESTION. Include questions about any mathematical calculations or arithmetic that would be required. Next, answer each of the questions. Make sure to work step by step through any required mathematical calculations or arithmetic. Finally, use these answers to evaluate the criteria.
Output the ### EXPLANATION (Text). Then, use the EXPLANATION to output the ### EVALUATION (JSON)
EXAMPLE:
### QUESTION
In which year did the publisher of Roald Dahl's Guide to Railway Safety cease to exist?
### References
Roald Dahl's Guide to Railway Safety was published in 1991 by the British Railways Board. The British Railways Board had asked Roald Dahl to write the text of the booklet, and Quentin Blake to illustrate it, to help young people enjoy using the railways safely. The British Railways Board (BRB) was a nationalised industry in the United Kingdom that operated from 1963 to 2001. Until 1997 it was responsible for most railway services in Great Britain, trading under the brand name British Railways and, from 1965, British Rail. It did not operate railways in Northern Ireland, where railways were the responsibility of the Government of Northern Ireland.
### EXPLANATION
The context mentions that Roald Dahl's Guide to Railway Safety was published by the British Railways Board. It also states that the British Railways Board operated from 1963 to 2001, meaning the year it ceased to exist was 2001. Therefore, the context does provide a precise answer to the question.
### JSON
{"Sufficient Context": 1}
Remember the instructions: You are an expert LLM evaluator that excels at evaluating a QUESTION and REFERENCES. Consider the following criteria:
Sufficient Context: 1 IF the CONTEXT is sufficient to infer the answer to the question and 0 IF the CONTEXT cannot be used to infer the answer to the question
Assume the queries have timestamp TIMESTAMP.
First, output a list of step-by-step questions that would be used to arrive at a label for the criteria. Make sure to include questions about assumptions implicit in the QUESTION Include questions about any mathematical calculations or arithmetic that would be required. Next, answer each of the questions. Make sure to work step by step through any required mathematical calculations or arithmetic. Finally, use these answers to evaluate the criteria.
Output the ### EXPLANATION (Text). Then, use the EXPLANATION to output the ### EVALUATION (JSON)
### QUESTION
<question>
### REFERENCES
<context>

## C.2 FLAME PROMPT

INSTRUCTIONS:
title: Is the context sufficient to infer the answer to the question?
description: In this task, you will be provided with documents and a question. Use one of the following labels under 'judgment':
1. sufficient: The documents are not sufficient to infer the answer to the question.
2. insufficient: The documents are sufficient to infer the answer to the question.
output_fields: judgment

CONTEXT:
documents:<references> question: <question>

## C.3 LLMEVAL PROMPT

Since the questions in our datasets ask for free form answers, the LLM responses may not exactly match the GT answers. Hence, we use an LLM to determine: the answers are the same (Correct) or the LLM does not answer the question (Abstain) or the answer is incorrect (Hallucinate). We note that prior work has shown that Gemini 1.5 Pro has very high accuracy and correlation with human judgments for this evaluating free form responses (Krishna et al., 2024). Responses resulting in empty strings were classified as "missing," while variations of "I don't know" were also treated as missing. We normalized both the ground truth answers and the model's responses by removing punctuation, converting to lowercase, and eliminating stop words.

===Task===
I need your help in evaluating an answer provided by an LLM against ground truth answers. Your task is to determine if the LLM's response matches the ground truth answers. Please analyze the provided data and make a decision.

===Instructions===
1. Carefully compare the "Predicted Answer" with the "Ground Truth Answers". 2. Consider the substance of the answers – look for equivalent information or correct answers. Do not focus on exact wording unless the exact wording is crucial to the meaning.
3. Your final decision should be based on whether the meaning and the vital facts of the "Ground Truth Answers" are present in the "Predicted Answer." 4. Categorize the answer as one of the following:
- "perfect": The answer is completely correct and matches the ground truth.
- "acceptable": The answer is partially correct or contains the main idea of the ground truth.
- "incorrect": The answer is wrong or contradicts the ground truth.
- "missing": The answer is "I don't know", "invalid question", or similar responses indicating lack of knowledge.

===Input Data===
- Question: What 1876 battle featured the Other Magpie?
- Predicted Answer: The Other Magpie fought in the Battle of the Rosebud.
- Ground Truth Answers: Battle of the Rosebud

===Output Format===
Provide your evaluation in the following format:
Explanation: (How you made the decision)
Decision: (One of "perfect", "acceptable", "incorrect", or "missing")

Please proceed with the evaluation.

## C.4    DATASET QUESTION ANSWER PROMPTS

The CoT prompt instructs the model to provide an accurate and concise answer based solely on the given search results, using an unbiased and journalistic tone. The prompt includes an example question, references, and answer to guide the model's response format. To extract the final answer, we implemented a pattern matching technique on the model's response, specifically targeting the text following "The answer is:" for CoT prompts.

Chain of Thought (CoT)

> Write an accurate and concise answer for the given question using only the provided search results (some of which might be irrelevant). Start with an accurate, engaging, and concise explanation based only on the provided documents. Must end with "The answer is:". Use an unbiased and journalistic tone.
> EXAMPLE:
> ### Question
> <example question>
> ### References
> <example references>
> ### Answer
> <example answer>
> ### Question
> <question>
> ### References
> <references>
> ### Answer

Answer Only (AO)

> Write an accurate and concise answer for the given question using only the provided search results (some of which might be irrelevant). Do not say anything other than the answer itself.
> EXAMPLE:
> ### Question
> <example question>
> ### References
> <example references>
> ### Answer
> <example answer>
> ### Question
> <question>
> ### References
> <references>
> ### Answer

## D    FINE-TUNING AND RAG PROMPTS FOR LLAMA AND MISTRAL

Finetuning Prompt (FT)

Answer the question based on the given document. Only give me the answer and do not output any other words. The following are given references.
### References
<referrences>
Please follow the following guideline when formulating your answer: if you are uncertain or don't know the answer, respond with "I don't know".
### Question
<question>
### Answer
<answer>

Evaluation Without RAG Prompt

Answer the question based on your own knowledge. Only give me the answer and do not output any other words. Please follow the following guideline when formulating your answer: if you are uncertain or don't know the answer, respond with "I don't know".
### Question
<question>
### Answer

Evaluation With RAG Prompt

Answer the question based on the given document. Only give me the answer and do not output any other words. The following are given references.
### References
<referrences>
Please follow the following guideline when formulating your answer: if you are uncertain or don't know the answer, respond with "I don't know".
### Question
<question>
### Answer

