# OpenReview forum: "Sufficient Context: A New Lens on Retrieval Augmented Generation Systems"
_ICLR.cc/2025/Conference — ICLR 2025 Poster_

### Official Review · Reviewer_BUto · 2024-10-26

**Soundness:** 3
**Presentation:** 3
**Contribution:** 3
**Rating:** 8
**Confidence:** 3

**Summary:**

This paper introduces a novel framework for understanding context sufficiency in Retrieval-Augmented Generation (RAG) systems, where retrieval of external context aims to improve Large Language Model (LLM) accuracy and reliability. The authors define “sufficient context” as context providing enough information for a model to answer a query without additional details. They introduce a sufficient context autorater to classify instances based on this metric, allowing for systematic error analysis across several proprietary and open-source LLMs. Key findings suggest that proprietary models (e.g., Gemini, GPT) often generate incorrect answers without abstaining when context is insufficient, while open-source models are more likely to hallucinate or abstain even when context suffices. The authors propose a selective generation technique to reduce hallucinations by leveraging sufficient context signals, showing improvements of up to 10% in response accuracy among Gemini, GPT, and Gemma.

**Strengths:**

1. Novelty of Sufficient Context Concept: The paper addresses a fundamental gap in RAG literature by formalizing the concept of “sufficient context.” This is essential for evaluating how well LLMs make use of retrieved information, especially when distinguishing between sufficiency and insufficiency.

2. Methodology and Rigorous Analysis: By creating a sufficient context autorater and testing it across proprietary and open-source models, the paper presents a solid empirical foundation. The detailed experiments on multi-hop and open-book QA datasets (e.g., Musique-Ans, HotPotQA, FreshQA) are impressive, particularly in showing how context sufficiency affects model performance.

3. Impact on Hallucination Mitigation: The proposed selective generation framework that combines sufficient context labels with model confidence offers a practical approach to reduce hallucinations. This approach is adaptable across different LLM architectures, which is an important contribution for model deployment.

4. Evaluation Metrics and Generalization: The paper extensively evaluates its autorater and selective generation framework across various datasets and models, which provides strong evidence of generalizability. The careful selection of open and proprietary models, and focus on multi-hop reasoning tasks, enhance the robustness of the conclusions.

**Weaknesses:**

1. Clarification on the Autorater’s Design and Reliability: The paper could better explain the autorater’s design, particularly regarding criteria used in challenging multi-hop contexts. Additional qualitative examples could help clarify how the autorater discerns sufficiency, especially when dealing with ambiguous or incomplete context.

2. Limitations in Practical Deployment: While the selective generation framework demonstrates significant accuracy improvements, there is limited discussion on computational overhead. For example, implementing a selective generation method based on sufficient context may require high processing power for real-time applications, particularly with large models and datasets.

3. Impact of Retrieval Quality: Although the paper acknowledges that retrieval quality can affect context sufficiency, the experiments do not address how different retrieval strategies impact results. Comparing RAG performance using advanced retrieval techniques, like iterative or corrective retrieval, would strengthen the argument that improving sufficiency is an independent aspect of model improvement.

4. Scope Beyond QA Tasks: The paper focuses on QA, but RAG systems are used across various tasks, including summarization, which the authors briefly mention. A future direction could involve validating sufficient context’s relevance to other NLP tasks, which would enhance the paper’s broader applicability.

5. Explain Model-Specific Findings in More Detail: The differential behavior of models, such as Gemini’s reduced abstention rate and GPT’s lower accuracy without RAG, would benefit from additional discussion. These findings highlight variations in model handling of retrieval, and more insight here could guide further model-specific improvements.

6. Consideration of Confidence Thresholds: The selective generation approach could further explore confidence thresholds to find an optimal balance between abstention and accuracy. Including ablation studies or sensitivity analyses on confidence levels would make this method more practical for different application contexts.

**Questions:**

1. How is "sufficient context" defined for multi-hop reasoning?
Clarify the specific criteria or thresholds the autorater uses, especially for complex multi-hop queries.

2. How does the autorater handle ambiguous or conflicting contexts?
What rules are applied when contexts contain contradictory or unclear information?

3. How does retrieval quality impact context sufficiency and model performance?
Could improved retrieval reduce hallucination rates by providing more sufficient contexts?

4. What are the computational demands of the selective generation framework?
Is it feasible for real-time applications, particularly with large models?

5. Is the sufficient context framework applicable beyond QA?
Could this framework be useful for other retrieval-augmented tasks, like summarization?

---

> ### Author Response · Authors · 2024-11-19
>
> Thank you for your time, feedback, and positive review! We respond to your comments and questions below. To answer your questions, we have included new discussions about our autorater and its broad applicability for both practical systems and research endeavors.
>
> > How is "sufficient context" defined for multi-hop reasoning? Clarify the specific criteria or thresholds the autorater uses, especially for complex multi-hop queries.
>
> The criteria for multi-hop is the same as for single-hop. The context must have enough information to answer the question. In the multi-hop setting, the context needs to have answers to the intermediate questions when it is necessary. Since the main autorater we use is a prompted version of Gemini, we do not need to include precise rules to differentiate between single-hop and multi-hop questions.
>
> > How does the autorater handle ambiguous or conflicting contexts? What rules are applied when contexts contain contradictory or unclear information?
>
> This is a very good question. The case of contradictory information requires handling on a case-by-case basis. For example, if the context contains multiple valid answers to the question, then this is the “ambiguous context” case, and the context is sufficient. If the context contains unclear information, where the question is not definitively answerable, then it is insufficient.
>
> > How does retrieval quality impact context sufficiency and model performance? Could improved retrieval reduce hallucination rates by providing more sufficient contexts?
>
> Yes, absolutely. This is one of the main takeaways from our work, coupled with the issue of models hallucinating even with sufficient context. At a high level, the best solution to improve RAG performance is to develop a way to get a very high percentage of sufficient context. The models all have a higher % of correct responses with sufficient context. Nonetheless, another important finding is that models hallucinate 10–26% of the time even with sufficient context. Hence, even perfect retrieval will not lead to completely factual answers! This implies that there is more work to be done around pre-training or fine-tuning LLMs to improve the utilization of the retrieved contexts.
>
> > What are the computational demands of the selective generation framework? Is it feasible for real-time applications, particularly with large models?
>
> Yes, it is indeed feasible to incorporate with very large models! The two components, model confidence (i.e., P(Correct)) and the sufficient context label are both quite efficient. To get P(Correct), we can use the same generation step as the answer generation, where we also get a confidence score along the way. To get a sufficient context label, we use the FLAMe 24B model, which is significantly more efficient and cheaper to run than the largest LLMs.
>
> > Is the sufficient context framework applicable beyond QA? Could this framework be useful for other retrieval-augmented tasks, like summarization?
>
> Yes, it is definitely applicable beyond QA. We are working on applications now as a follow-up work. Summarization is a good candidate. Another option is search queries, which are information-seeking but may not be fully formulated questions. A final use case is as a re-ranking signal after retrieving a large number of supporting documents. Overall, we believe sufficient context and follow-up ideas will be instrumental in shaping the future of RAG research, for academic and enterprise applications!

---

> > ### Author Response · Authors · 2024-11-22
> >
> > Dear Reviewer,
> >
> > Thank you again for your positive review! We are pleased to hear that you found that _"The paper extensively evaluates its autorater and selective generation framework across various datasets and models, which provides strong evidence of generalizability. The careful selection of open and proprietary models, and focus on multi-hop reasoning tasks, enhance the robustness of the conclusions."_
> >
> > We hope that we have addressed your interesting questions through our responses above. As the discussion period is coming to an end soon, please do not hesitate to share any remaining comments you may have.
> >
> > Cheers,
> >
> > Authors

---

> > ### Comment · Reviewer_BUto · 2024-11-27
> >
> > Thank you for providing a thorough and thoughtful response. I appreciate the time and effort you have dedicated to addressing the concerns and suggestions raised. I look forward to the follow-up work.

---

> ### Comment · Area_Chair_Kdix · 2024-11-25
> **Reminder: Rebuttal Deadline for ICLR 2025**
>
> Dear Reviewer BUto,
>
> As the rebuttal deadline approaches, please kindly check the papers' discussion threads and respond to the authors' rebuttals. If you haven't had a chance to respond yet, I’d greatly appreciate your input soon. Your insights are invaluable to the authors and the review process.
>
> Thank you for your effort and support!
>
> Best regards,
>
> Area chair

---

### Official Review · Reviewer_pMDo · 2024-10-27

**Soundness:** 3
**Presentation:** 2
**Contribution:** 1
**Rating:** 3
**Confidence:** 3

**Summary:**

The paper studies the performance of LLMs with retrieval-augmented context. In particular, the paper analyses how often the retrieved context is sufficient for answering a question. It then analyses how the behaviour of LLMs changes depending on whether the retrieved context is sufficient or not, i.e. in how many cases the LLM hallucinates, abstains, or answers correctly. Furthermore, the paper shows that checking for sufficiency can be used to "teach" an LLM to abstain (either by directly using the sufficiency prediction in combination with LLM-generated confidence metrics, or by fine-tuning a model to abstain).

**Strengths:**

Retrieval-augmentation is a popular and promising method to improve the performance of LLMs. Previous work has shown, however, that there are issues with this approach, especially if the retrieved context is not helpful and/or too long. The analysis in the paper contributes to the study of these effects.

The paper is easy to follow and some of the experimental results may be useful to some practitioners.

**Weaknesses:**

As the paper acknowledges in the related work section, there is a substantial amount of work that has already analysed the performance of RAG systems in relation to the quality of the retrieved context. The paper claims to study this topic through "a new lens", but the specific aims of the paper are not really clear. What is it really that this paper shows that wasn't already known? Perhaps a more explicit formulation of hypotheses/research questions that have not yet been answered could help here. In its current form, the paper reads like a preliminary exploratory analysis of LLM performance, rather than a mature contribution that is ready for publication.

The notion of "sufficient context" is central to this paper, but it is not actually defined in a very precise way. First, the definition (line 157) is rather "circular": to determine if a context is sufficient, we need to determine if it allows us to infer the answer, but that actually depends on the capabilities of the LLM. The definition is also confusing, as it talks about "plausible answers" rather than the actual answer. I think what is meant is that a context can be sufficient even if the answer it contains is incorrect. In any case, this point needs a lot more discussion. There are further issues, for instance, the definition assumes that an answer can be inferred up to 4 inference hops. Why 4?

To assess which contexts are sufficient, the paper relies on LLM-generated assessments. This is a key limitation, and the current analysis should be expanded with a human-annotated experiment. Indeed, currently, the paper does not actually study the impact of contexts being sufficient, but rather the impact of an LLM thinking that the context is sufficient. Even if the LLM that predicts sufficiency is different from the one used for generating the answer, this kind of dependency on LLMs may clearly affect the results.

There are lots of minor issues with the writing and with formulations being imprecise.

* The abstract contrasts the behaviour of closed models with open-source ones. But isn't the contrast rather between larger and smaller models?
* The example in Remark 2 is of an ambiguous context rather than an ambiguous query.
* The example in Remark 3 is of an ambiguous query rather than an ambiguous context
* Lines 225-228: where are these results shown?
* Table 2: I believe the explanation on the first row should be "some chance" and the explanation on the second row should be "50% chance"
* Line 384: sufficient -> insufficient

**Questions:**

What can we really conclude without manual validation of the auto-rater predictions?

Is the difference between context sufficiency and NLI based methods really that significant (for practical purposes)?

What are the key insights that your analysis has revealed that weren't known from previous work?

---

> ### Author Response · Authors · 2024-11-19
>
> Thank you for your time and feedback! We respond to your comments and questions below. To answer your questions, we have (a) clarified our research questions, (b) addressed a few potential misunderstandings about our definition and autorater, and (c) included new experiments.
>
> > What is it really that this paper shows that wasn't already known? Perhaps a more explicit formulation of hypotheses/research questions that have not yet been answered could help here.
>
> Our work starts with the overarching question:
>
>
> 1. Can we develop a principled and scalable way to analyze RAG responses in a way that separates retrieval issues from response generation issues?
>
>
> Then we move on to
>
>
> 2. How can we use insights from (1) to improve generation in RAG systems?
>
>
> To address (1), we first develop and validate a new autorater for classifying sufficient context. As the ground truth may not be available in production, we focus on a signal that only depends on the question and context. Our autorater enables a scalable way to investigate retrieval failures across multiple datasets and models (without relying on ground truth answers).
> Using our autorater, we find new trends: (a) SOTA LLMs hallucinate even with sufficient context, (b) LLMs hallucinate rather than abstain with insufficient context, (c) sufficient context auto-rater labels can improve selective generation.
> For (2), we use our analysis in Section 4 to inform the interventions and improvements we propose in Section 5. Overall, we believe our sufficient context idea and intervention methods will resonate with engineers who build enterprise RAG systems. The distinction between relevance and sufficient context is subtle but integral to building factual generative search.
> > To assess which contexts are sufficient, the paper relies on LLM-generated assessments. This is a key limitation, and the current analysis should be expanded with a human-annotated experiment . . . What can we really conclude without manual validation of the auto-rater predictions?
>
>
> We include a human-annotated experiment to validate the autorater in Table 1, where the LLM-based autorater is 93% accurate on representative examples. Overall, your question highlights the broader challenge of relying on LLM-based labeling versus human labels to analyze LLM behavior. We agree that this approach merits caution – it is important to verify that our observations are not just artifacts of auto-rater labeling. In this work, we can make conclusions without manual validation because the insights in the paper are robust to a small % change in the sufficient context labels.
>
>
> Based on your comment, we conducted new experiments about model accuracy on data with human-labeled sufficient context labels – please see the tables and discussion in our overall response to all reviewers.
>
>
> > The notion of "sufficient context" is central to this paper, but it is not actually defined in a very precise way. First, the definition (line 157) is rather "circular": to determine if a context is sufficient, we need to determine if it allows us to infer the answer, but that actually depends on the capabilities of the LLM.
>
>
> We want to clarify this point, as we think this may reflect a misunderstanding. The definition of sufficient context we put forth is independent of the ability of the LLM that answers the question. We aim to rate questions in a way that applies to LLMs, humans, and any other systems that use retrieval to answer a question.
> > The definition is also confusing, as it talks about "plausible answers" rather than the actual answer. I think what is meant is that a context can be sufficient even if the answer it contains is incorrect.
>
>
> Your understanding here is accurate – the context can be sufficient even if it contains an “incorrect” answer. This is a key requirement, because (i) we would like to be able to use signal from sufficient context at inference time, where we do not have ground truth answers (see Section 5.1) and (ii) we hope to elucidate findings that are robust to ground truth label noise.
> > Is the difference between context sufficiency and NLI based methods really that significant (for practical purposes)?
>
>
> The core difference is that NLI-based methods need a ground truth answer. Hence, they do not apply to on-the-fly analysis for RAG systems. This is critical for using sufficient context at inference time, as in Section 5.1, or when the ground truth is not available, as in production systems.
> > The definition assumes that an answer can be inferred up to 4 inference hops. Why 4?
>
>
> Thank you for raising this. While this may seem arbitrary, existing RAG benchmarks have questions with 2–4 inference hops by design (e.g., Musique, HotPotQA). Hence, we limit the scope of sufficient context to match these. We will clarify this in the updated version.
>
> > minor issues with the writing and with formulations
>
>
> Thank you for your careful reading and for flagging these issues! We’ve corrected these.

---

> ### Author Response · Authors · 2024-11-19
>
> > What are the key insights that your analysis has revealed that weren't known from previous work?
>
> One of the key insights is that SOTA LLMs tend to hallucinate rather than abstain when given insufficient context. This was not known before, given that prior work did not distinguish between sufficient context with distracting information and insufficient context. This insight also informed our approach for selective generation (Section 5.1) and fine-tuning (Section 5.2), where we cannot just use sufficiency as a signal by itself for the model to answer or abstain. Another insight, as we mentioned above, is that models hallucinate on a non-trivial fraction of examples with sufficient context.
> Since you mention prior work, we want to point out that previous papers focus more on robustness to distracting content rather than distinguishing model behavior based on context sufficiency.
> For example,
> * Xie et al (ICLR 2024) show that “LLMs can be highly receptive to external evidence even when that conflicts with their parametric memory.” which is a different goal from our paper.
> * Wu et al (COLM 2024) show that “current LLMs still face challenges in discriminating highly semantically related information…. current solutions for handling irrelevant information have limitations in improving the robustness of LLMs to such distractions” which, in our language, is a study of robustness to different types of insufficient context.
> * Yoran et al (ICLR 2024) “propose a method for automatically generating data to fine-tune the language model to properly leverage retrieved passages, using a mix of relevant and irrelevant contexts at training time.” Their source of relevant vs. irrelevant contexts involves high-ranked vs. low-ranked search results, and hence, this does not accurately separate sufficient vs. insufficient context.
>
> In general, compared to these works, we add new insights about RAG system analysis, including the breakdown into accurate, abstain, and hallucinate, as opposed to just correct vs. incorrect. It is possible we failed to find or cite highly related work in our paper – if you find this to be the case please let us know!

---

> > ### Comment · Reviewer_pMDo · 2024-11-20
> >
> > Thanks for your clarification about the analysis with ground truth labels in Table 1. The FLAMe model, which is used for the subsequent analysis, has an accuracy of 87.8%. While this is of course very reasonable, it doesn’t completely alleviate my concerns about the impact of using these LLM-based assessments of sufficiency for the analysis. The model still makes mistakes, and this is likely to have an impact on the analysis. For instance, we don’t know how models behave with contexts that are sufficient but where the LLM thinks otherwise.
> >
> > Regarding the similarity with NLI, the point is that you could apply NLI on the context and the LLM generated answer. If the LLM-generated answer is entailed by the context, then you consider the context sufficient. This seems, conceptually, rather similar to what is proposed in the paper. This NLI view is adopted, for instance, in the RAGAS evaluation framework (https://docs.ragas.io/en/stable/concepts/metrics/available_metrics/faithfulness/), where “sufficiency” is called faithfulness.
> >
> > Thanks for the careful summary of how the paper differs from previous works. I accept that your analysis is somewhat different from what has been done in the literature. But I’m still not entirely convinced of the significance of this analysis in light of what we already know about LLMs. We obviously know that LLMs hallucinate and we know that LLMs get distracted by irrelevant context. The idea of improving RAG systems by checking whether the answer is entailed by the context makes sense, but similar ideas have already been considered (for instance by RAGAS).

---

> > > ### Author Response · Authors · 2024-11-21
> > > **Author Response to Follow-up Questions**
> > >
> > > Thank you for your continued discussion and for the great follow-up questions. At a high level, we are both working toward the same goals: presenting a valid analysis while also properly comparing our work to prior papers. It seems there also might still be some misunderstandings about the autorater and NLI-based approach, which we address below. Overall, if our added clarifications alleviate your concerns, please consider raising your score. If not, please let us know what else we can help with (we have tried to be as clear as possible in order to be respectful of your time).
> > >
> > > > The model still makes mistakes, and this is likely to have an impact on the analysis.
> > >
> > > Based on your questions, we present in [Tables 1 and 2 in the overall response](https://openreview.net/forum?id=Jjr2Odj8DJ&noteId=6qWsUaT9ma) **a new experiment where we use human-annotated sufficient context labels for the analysis**. We validate that our conclusions also hold in this case (and we agree that such validation is a crucial component of our study). Please see those tables and our response to all reviewers for more details. Hence, we believe that using the LLM as an autorater is a feasible way to scale this analysis to more models and larger datasets.
> > >
> > > > If the LLM-generated answer is entailed by the context, then you consider the context sufficient. This seems, conceptually, rather similar to what is proposed in the paper. This NLI view is adopted, for instance, in the RAGAS evaluation framework
> > >
> > > Thank you for raising this point. **There is actually a fundamental issue with using NLI on the response as a sufficient context autorater: systematic false negatives.** We should have been more clear about this in the paper, but we partially address it in lines 219 – 224:
> > >
> > > _“...if a given answer A is entailed by (Q,C), then the context is also sufficient. On the other hand, the reverse is not true…”_
> > >
> > > If we understand your proposal correctly, then the NLI-based approach on input (question, context, response) will _always_ output “insufficient context” when the LLM’s response is not entailed by the context. For example, if the LLM says “I don’t know” then this will not be entailed. Another case is when the LLM hallucinates an answer that is not in the context (e.g., if in the RAGAS example, the LLM happened to output "Einstein was born in Austria."). **This means the NLI-based approach will generate a large number of false negatives, where the context is actually sufficient but it is labeled as insufficient.**
> > >
> > > This matters because as you have noted, many mislabeled contexts would impact the analysis. For example, every time the LLM abstains, the NLI-on-the-response method would mark the context as insufficient. So, we would be led to believe that the model never abstains on sufficient context, which would be false.
> > >
> > > > I’m still not entirely convinced of the significance of this analysis in light of what we already know about LLMs. We obviously know that LLMs hallucinate and we know that LLMs get distracted by irrelevant context. The idea of improving RAG systems by checking whether the answer is entailed by the context makes sense, but similar ideas have already been considered (for instance by RAGAS).
> > >
> > > Above, we argue that sufficient context is indeed different than NLI-based methods or faithfulness. Beyond this aspect, it seems that the RAGAS paper does not provide an analysis of LLM behavior as a function of the faithfulness score. If you have seen such an analysis, please let us know. Because of this, we agree with reviewers that the “analyses are genuinely insightful and offer a number of non-trivial findings” (HFFU) and “reports findings of unexpected model behavior with and without sufficient context, which are valuable to the community.” We think that it is an important and novel message that LLMs still hallucinate quite often _even with sufficient context_.
> > >
> > > We agree that prior work has found “LLMs get distracted by irrelevant context”. However, we have not seen a principled definition of irrelevant context for RAG systems. **There is a big difference, in our opinion, between “the context is sufficient but contains some parts that have nothing to do with the question” versus “nothing in the context provides the answer to the question”.** So, we believe that conflating these two types of context glosses over the more direct question of: _did the model make an error even though it should have been able to answer the question correctly?_
> > >
> > > > The FLAMe model, which is used for the subsequent analysis, has an accuracy of 87.8%.
> > >
> > > We would like to clarify that **the FLAMe model is not used for the analysis – we use the Gemini 1.5 Pro for the analysis (accuracy 93%)**. We only used FLAMe for the results in Section 5.1 on selective generation. We used FLAMe for selective generation experiments because they focus on methods that could be used in production, where the multiple calls to an LLM could be cost-prohibitive.

---

> > > > ### Comment · Reviewer_pMDo · 2024-11-22
> > > >
> > > > Thanks for the clarification about Tables 1 and 2 in the main response, I had indeed missed this.
> > > >
> > > > Regarding the differences with the NLI approach, I'm not convinced that this would matter in practice, or that there would indeed be a large number of false negatives.
> > > >
> > > > Overall, I still think the contribution is too incremental, so I won't be increasing my score.

---

> > > > > ### Author Response · Authors · 2024-11-22
> > > > >
> > > > > Thank you for the follow-up comment and for acknowledging the experiments we added in the rebuttal to directly address one of your main questions. We also appreciate that you have engaged in a back-and-forth discussion, making this a fruitful OpenReview process!
> > > > >
> > > > > We also thank you for the questions about the NLI-based approach. We had not thought deeply about the pitfalls of using NLI on the model response, and we can incorporate this justification for using sufficient context in the future. In Table 1 in the main paper, we used NLI on the ground truth answer, and found **the TRUE-NLI model recall was only 0.726 compared to a recall of 0.935 from Gemini 1.5 Pro (1-shot)**. As they have comparable True Positives, this is direct evidence for the increased False Negatives with NLI.
> > > > >
> > > > > We respect that you still find the work to be incremental. If you have any concrete suggestions for follow-up experiments for next version of our paper, we would be happy to hear them! We are actively using sufficient context as a signal in multiple RAG systems, so this would help us increase our impact.

---

> > > > > > ### Author Response · Authors · 2024-12-02
> > > > > >
> > > > > > Since this is the last day for author-reviewer discussion, we wanted to make sure you saw our comments above to reviewer 9Kme -- some of their concerns were directly based on your comments and our responses, so we we addressed these with additional experiments and qualitative analysis.
> > > > > >
> > > > > > Specifically, we added (see [here](https://openreview.net/forum?id=Jjr2Odj8DJ&noteId=6BnELoX4N0) and [here](https://openreview.net/forum?id=Jjr2Odj8DJ&noteId=3zSSaIGhGy)):
> > > > > > 1. **Clarifications** about how the change in correct % was due to the different dataset in the rebuttal, and it is not an issue with our evaluation set-up
> > > > > > 2. **Experiments** with a new deterministic metric ("Contains Answer") where these results verify that our main conclusions hold even if we remove the LLM QA Evaluator from the eval loop, showcasing robustness in our findings
> > > > > > 3. **Examples** showing the LLM QA Evaluator is better at handling certain model responses, where it labels the response better than the deterministic method, indicating that our evaluation pipeline is intentional and rigorous
> > > > > >
> > > > > > Overall, one of the valuable parts of our work is showing exactly how/when LLM-based evaluation methods can lead to a **scalable** and **multi-faceted** analysis of model responses. We believe that this will inspire researchers in the future to also **dive deeper into system-level performance** by utilizing autoraters to stratify datasets (after properly verifying their performance and checking for bias, as we do in our paper).

---

### Official Review · Reviewer_HFFU · 2024-11-04

**Soundness:** 3
**Presentation:** 3
**Contribution:** 3
**Rating:** 6
**Confidence:** 4

**Summary:**

This work asks why LLMs hallucinate when integrated into RAG systems. They carefully define "sufficient context", i.e. whether the retrieved context unambiguously supports some answer for a question, while making sure their definition is appropriate for various types of RAG systems, e.g. multi-hop reasoning. They then develop an "autorater" prompt for this quality, which they validate on a set of 115 hand-labeled and challenging data points. Using this autorater, they analyze the performance of an off-the-shelf retrieval system (REPLUG) across datasets and then analyze the performance of different LLMs in RAG, exploring their behavior (correct answers vs. abstaining vs. hallucination) when sufficient context is or isn't available. Overall, they find that models abstain less with RAG and hallucinate whether sufficient context is available or not. They build a small set of reasons answering questions correctly despite insufficient context, e.g. when it's due to models using parameteric knowledge when faced with incomplete multi-hop information. Leveraging their autorater, the authors explore a number of approaches for creating a tradeoff between tuning abstention, e.g. only answering when context is clearly sufficient and the system is confident in high-stakes applications, while relaxing that (to answer more questions) elsewhere.

**Strengths:**

1. The paper is well-written and easy to follow.
2. The analyses are genuinely insightful and offer a number of non-trivial findings, as included in my summary.
3. The autorater, its validation set, the analyses approach, and the method proposed for trading selectivity vs. coverage are all likely to be valuable resources for future work.

**Weaknesses:**

1. How sensitive is the analysis to the choice of REPLUG and/or E5 as retriever components? The paper glosses over the IR subsystem here in an extreme way. Most of this information is simply not presented. For example, I have to read the results in Figure 2 as essentially an evaluation of REPLUG, but they could also be read as (and appear to be presented as) analyses of the datasets themselves. Is REPLUG able to handle multi-hop queries? I am not aware that it's a multi-hop retriever at all, so the choices here, while still plausible, are lacking in justification.

2. How good is the autorater? The paper includes table 1 but to my knowledge does not directly refer to it; and overall, the paper doesn't seem to offer a convincing discussion of this, given how essential the autorater is to the analysis and methods. Will the 115 data points be released? There's some risk of circularity here. What makes large LLMs so prone to hallucination if one could trust that a simple prompt could reliably discover these cases?

3. Given that the autorater is not perfect, and given the subtle nature of its job (as discussed in the Remarks under the definition of Sufficient Context), how correlated are autorater mistakes with determinations of Correct/Hallucination in Figure 3?

**Questions:**

See Weaknesses.

---

> ### Author Response · Authors · 2024-11-19
>
> Thank you for your time and feedback! We respond to your comments and questions below. To answer your questions, we have (a) clarified how our work generalizes across IR subsystems, (b) addressed a few potential misunderstandings about our definition and autorater, and (c) included new experiments with human labels.
>
>
> > How sensitive is the analysis to the choice of REPLUG and/or E5 as retriever components? The paper glosses over the IR subsystem here in an extreme way.
>
>
> Thank you for this question – you raise an important point about how our work relates to research on information retrieval systems. We would argue that our paper generalizes beyond the IR subsystem rather than glosses over it. Concretely, we believe splitting into sufficient vs. insufficient context is a principled way to ablate the retriever since all retriever and re-ranker methods will produce a mix of sufficient and insufficient context. This is why we separately analyze both context types and derive insights into how LLMs respond in each case.
>
>
> We also want to clarify only HotPotQA uses the REPLUG retriever. For FreshQA, we use the first 6000 tokens of the oracle URLs. For Musique, we use the provided context in full (which has 20 paragraphs, some of which are relevant to the question).
>
>
> > How good is the autorater? … Given that the autorater is not perfect, and given the subtle nature of its job (as discussed in the Remarks under the definition of Sufficient Context), how correlated are autorater mistakes with determinations of Correct/Hallucination in Figure 3?
>
>
> This is a good question. We include a human-annotated experiment to validate the autorater in Table 1, where the LLM-based autorater is 93% accurate on representative examples. We also provide new experiments about model accuracy on data with human-labeled sufficient context labels in the overall response to all reviewers (see the added tables for Correct/Abstain/Hallucination). We corroborate our findings with this added experiment.
>
>
> Your question highlights the broader challenge of relying on LLM-based labeling versus human labels to analyze LLM behavior. We agree that this approach merits caution – it is important to verify that our observations are not just artifacts of the auto-rater labeling. In this work, we can make conclusions without manual validation because the insights in the paper are robust to a small % change in the sufficient context labels. Even if the LLM autorater misclassified some examples as sufficient vs. insufficient, the findings still hold.
>
>
> > Will the 115 data points be released?
>
>
> We would like to release the data, assuming we can get approval to release the dataset or a similar version. The challenge is that the context contains snippets taken from a variety of sources, which makes licensing complicated. As a workaround, we are currently working on an experiment to human-label a large subset of questions from Musique to verify the performance of the autorater, but this experiment may not finish before the end of the discussion period.
>
>
> > There's some risk of circularity here. What makes large LLMs so prone to hallucination if one could trust that a simple prompt could reliably discover these cases?
>
>
> Thank you for this question. We want to clarify that we do not believe there is circularity here, but rather we discover a new disparity between LLM capabilities as an autorater vs. as a generation model. Indeed, we believe that the sufficient context autorater hallucinates less than the question-answering LLMs. Given that they are different tasks, there is no immediate contradiction here. Moreover, this disparity of capabilities can hold true for humans as well. It is believable that a person could achieve high accuracy on the binary labeling task of sufficient vs. insufficient context. But the same person may have lower accuracy in a free-response test that requires filling in the answers to questions.
>
>
> Overall, it is a wonderful, and deep, question to ask “What makes large LLMs so prone to hallucination?” With a grain of salt, our intuition is that the model doesn’t have a good internal mechanism to decide whether to abstain or to answer the given question. Hence, the model starts to generate a response based on the context, even when it is insufficient.
>
>
> Section 5 in our paper discusses ways to improve RAG systems by capturing the impressive capability of LLMs to rate sufficient context. For example, we show in Section 5.1 that we can do better with selective generation by using confidence and sufficient context as signals to decide whether to abstain or answer. In Section 5.2, we use the sufficient context labels to modify the fine-tuning data mixtures.

---

> > ### Author Response · Authors · 2024-11-22
> >
> > Dear Reviewer,
> >
> > Thank you again for your positive review! We are pleased you found that _"The analyses are genuinely insightful and offer a number of non-trivial findings."_
> >
> > We hope that we have addressed your insightful questions through our responses above. As the discussion period is coming to an end soon, please do not hesitate to share any remaining comments you may have.
> >
> > Cheers,
> >
> > Authors

---

> > > ### Comment · Reviewer_HFFU · 2024-11-25
> > >
> > > Thanks for the response. I'll keep my score.

---

### Official Review · Reviewer_9Kme · 2024-11-06

**Soundness:** 3
**Presentation:** 3
**Contribution:** 3
**Rating:** 8
**Confidence:** 3

**Summary:**

This work analyzes how LLM behavior differs for question answering with and without sufficient context. A context C is considered sufficient to answer a question Q if it supports an answer A in C. The authors make it a point to ensure that the ground truth answer does not matter in the definition of sufficient context.

First, the authors collect an evaluation set of 115 (question, context, answer) triplets (from 3 QA datasets) and annotate them for sufficiency. A set of autoraters is evaluated on the task of judging sufficiency. Gemini 1.5 Pro (1-shot) achieves an F1 score of 93% on evaluating sufficiency. For the rest of the analysis, Gemini 1.5 Pro (1-shot) labels examples as having sufficient/insufficient context.

Next, the paper analyzes how context sufficiency differs for different QA datasets. (Question, context) triples are collected for 3 popular QA datasets: FreshQA, HotpotQA, and Musique. Contexts are either gold snippets from the dataset (for Musique) or retrieved with a dense retriever. Results show that sufficiency does not increase significantly after the first 6000 tokens of context. FreshQA has the most sufficient per query while the context is sufficient for only about 45% of the questions from Musique and HotPotQA.

Next, the paper analyzes how context sufficiency affects the accuracy of Retrieval Augmented Generation systems. They study the behavior of 4 API-based QA models.
1. The accuracy of QA models is higher when given sufficient context than without. However, even without sufficient context, the models answer the query about 35% of the time.
2. When given sufficient context, models have a low abstention rate and relatively higher error rate.
3. When given insufficient context, the abstention and hallucination (error) rates both increase.

The authors show a qualitative breakdown of the questions the QA models answer correctly with insufficient context.

Finally, the paper studies ways to use the "context sufficiency predictor" for a well-calibrated QA model. The sufficiency predictor can be used as a signal for selective prediction. This provides some gains over using just the self-reported model confidence especially with a worse QA model. Fine-tuning the QA models to abstain (either on random contexts or contexts predicted to be insufficient) improves the ability to abstain but does not improve overall QA accuracy.

---

Scores changed from 8 -> 6. See reasoning [here](https://openreview.net/forum?id=Jjr2Odj8DJ&noteId=fVAH86Vrwz) and [here](https://openreview.net/forum?id=Jjr2Odj8DJ&noteId=a0bQQDrq6s).

---

Final Update After Rebuttal
---

I'm satisfied with the author responses. I am changing my score back to 8. Ideally, this comes with the caveat: I think the core argument of the paper and findings have value, but I believe that the final version of the paper needs detailed discussions based on the author-reviewer discussion.
- Clear discussions of the different QA benchmarks and their impact. This will address issues of why different findings are not directly comparable and the nuances of why certain QA datasets behave differently.
- The final version of the paper should include a **quantitative count** of the different ways the QA models are correct given insufficient context. This would also better answer questions about automated metric reliability (rather than anecdotal evidence)

**Strengths:**

- The presentation of the paper is clear and easy to follow.
- Appropriate evaluations are used at each stage. The sufficiency predictor is first evaluated and then used for later analysis.
- The paper reports findings of unexpected model behavior with and without sufficient context, which are valuable to the community.
    - It is interesting that Gemini 1.5 Pro does not abstain very often even on passages that Gemini 1.5 Pro has labeled has having insufficient evidence (as autorater)

**Weaknesses:**

1. Some details in the paper are vague. While they don't obstruct understanding, they may obstruct reproducibility [see Questions]
2. While the paper spends a lot of time analyzing the case of correct answer with insufficient context, the opposite case of incorrect answer with sufficient context is equally interesting and warrants exploration
3. The qualitative evaluation does not attempt to quantify any of the error types. For example, how prevalent is the "Autorater error" [see Questions]

**Questions:**

1. The autorater is evaluated on single-hop QA datasets like PopQA, FreshQA, Natural Questions, and EntityQuestions. The rest of the paper analyzes QA performance on the FreshQA (in-domain) and potentially out-of-domain HotpotQA and Musique datasets. Can the automatic judgments be trusted out-of-domain?
2. Following up: since you observe rater error in Table 2, what is the estimated prevalence of this category?
3. One other potential mechanism of the correct answer given insufficient context is that there are only one/few entities of the expected type (given the question) in the context. Is this mechanism never observed?
4. What is the context source for FreshQA? Is it the labeled sources? If so, are there no examples of insufficient sources for FreshQA questions in the automated evaluation dataset?
5. Lines 185-190: Remark about ambiguous queries: It is unclear to me why the given example illustrates how ambiguous queries are handled. My understanding is that ambiguous queries have unclear interpretations, not the context.
6. Do you plan to release the sufficient context dataset? If not, how do you ensure the following: "very challenging, including single- and multi-hop questions, as well as adding highly related information in the context even if it is not sufficient"

---

> ### Author Response · Authors · 2024-11-19
>
> Thank you for your review and positive characterization of our work! We respond to your comments and questions below. To answer your questions, we have (a) clarified details about our autorater and error analysis, (b) addressed your comments about dataset release, and (c) included new experiments.
>
> > Can the automatic judgments be trusted out-of-domain?
>
>
> This is a great question! Out of the box, the prompt we use for sufficient seems to work well for open-book QA questions. We have manually verified dozens of examples from FreshQA, Musique, and HotPotQA and found the accuracy to be consistent with the results in Table 1. We are also working on an experiment to human-label a large subset of questions from Musique to verify the performance of the autorater, but this experiment may not finish before the end of the discussion period. We have also looked at many more out-of-domain questions, including search queries or enterprise RAG systems (e.g., internal documents), and we find that small variations in the prompt usually suffice to achieve good performance here as well.
>
> > Since you observe rater error in Table 2, what is the estimated prevalence of this category?
>
> In this category we have grouped together two sources of errors: (1) For the sufficient context autorater, the error rate is about 7%, which seems consistent across both the human-labeled data and the other datasets. (2) Another error rate is when the LLM Eval labels an answer as `Correct` when instead it should be `Abstain` or `Hallucinate`. Our observation is that this error is rare, and other papers (e.g., Krishna et al) find it to be 0.5% based on human quality checking. We have also taken care that our main conclusions remain valid even accounting for the prevalence of these two types of errors.
>
> > One other potential mechanism of the correct answer given insufficient context is that there are only one/few entities of the expected type (given the question) in the context. Is this mechanism never observed?
>
> This is a good point. When we have observed this, it falls into the category of “Multi-hop: fragment” where the model assumes that the entity of the expected type is also the answer to the question by inferring an extra reasoning step to connect the dots. We will look more closely at the data and add another row to the table if this mechanism is observed in more cases.
>
> > What is the context source for FreshQA? Are there no examples of insufficient sources for FreshQA questions in the automated evaluation dataset?
>
> Yes, the context sources for FreshQA are “oracle” URLs that support the answer. However, in some cases these sources are not complete, leading to insufficient context. This happens because the sources change or because the first 6k tokens in the source do not suffice to answer the question. For the automated evaluation dataset, we also included some examples where we manually modified the context where we deleted or rephrased the context to make it sufficient or insufficient. This also made the dataset roughly balanced.
>
> > While the paper spends a lot of time analyzing the case of correct answer with insufficient context, the opposite case of incorrect answer with sufficient context is equally interesting and warrants exploration
>
> We agree! We investigated cases where the autorater labels an instance as having sufficient context while the LLM evaluator marks the answer as incorrect. One source of these discrepancies occurs when the ground truth answer conflicts with the answer provided in the source. Another source of errors arises when the autorater correctly identifies that the necessary information is present, but the model fails to properly compose the information (e.g., in multihop questions or questions requiring arithmetic). In a substantial number of cases, however, determining the source of the error proves challenging. Thank you for expressing interest in this question – we’ve added commentary on this topic to the current version.
>
> > Do you plan to release the sufficient context dataset?
>
> We would like to, assuming we can get approval. The challenge is that the context contains snippets taken from a variety of sources, which makes licensing complicated. As a workaround, we are currently working on an experiment to human-label a large subset of questions from Musique to verify the performance of the autorater, but this experiment may not finish before the end of the discussion period.
>
> > Lines 185-190: Remark about ambiguous queries: It is unclear to me why the given example illustrates how ambiguous queries are handled.
>
> Thank you for pointing this out. The remark titles were swapped (e.g., the remark labeled ambiguous queries had the example for ambiguous context). We have updated this.

---

> ### Author Response · Authors · 2024-11-22
>
> Dear Reviewer,
>
> Thank you again for your positive review! We are pleased you found that _"The paper reports findings of unexpected model behavior with and without sufficient context, which are valuable to the community."_
>
> We hope that we have addressed your thought provoking questions through our responses above. As the discussion period is coming to an end soon, please do not hesitate to share any remaining comments you may have.
>
> Best,
> Authors

---

> ### Comment · Reviewer_9Kme · 2024-11-24
> **Response to rebuttal**
>
> Thank you for clarifying my questions.
>
> I have raised my **concern about the new results** under the common response. ~I believe that the discrepancy there is at odds with your comments about the reliability of the LLM QA evaluator.~ This discrepancy raises questions about the main results and the importance of the human analysis of "correct answers with insufficient context".
>
> **Re: Reliability of the LLM QA evaluator**: How does the QA evaluator behave when comparing against the ground truth vs an NLI-based evaluator (as suggested by Reviewer pMDo)? The NLI evaluator could serve as a way to judge the correctness of the LM-generated answer against the document context. This is different from the LLM QA evaluator because it would not be based on the ground truth answer. Looking at both evaluators may help to address the discrepancy.
>
> As such, I believe that the main message is very valuable: "judging sufficiency correctly does not translate to being able to abstain. But given the mismatch, I am decreasing my score to 6.

---

> > ### Author Response · Authors · 2024-11-28
> >
> > We are very sorry to hear that the additional experiments have led you to decrease your score to a 6. We would like to clarify some potential misunderstandings. We hope that we can work together to resolve this confusion, and ideally, get back to a place where we all agree on the high quality and rigor of our study. We also want to emphasize the novelty of using “sufficient context” to understand RAG performance, which should inspire future researchers to more deeply analyze LLM errors based on the provided context. We respond here as well as above in a reply to your comment on the new experiments.
> >
> > > I believe that the discrepancy there is at odds with your comments about the reliability of the LLM QA evaluator.
> >
> > This discrepancy is due to the different dataset used and not due to the reliability of the LLM QA evaluator. We comment on this in more detail in our other response.
> >
> > > How does the QA evaluator behave when comparing against the ground truth vs an NLI-based evaluator (as suggested by Reviewer pMDo)? The NLI evaluator could serve as a way to judge the correctness of the LM-generated answer against the document context. This is different from the LLM QA evaluator because it would not be based on the ground truth answer. Looking at both evaluators may help to address the discrepancy.
> >
> > First, we had a different understanding of the suggestion by Reviewer pMDo. We interpreted their comments as a proposal to use an NLI-based evaluator to label sufficient context (not to measure correctness). Using NLI to label sufficient context has many issues, such as mislabeling many contexts as insufficient when they are indeed sufficient (e.g. when the LLM abstains or hallucinates something not in the context).
> >
> > Second, if we understand your proposal correctly, then using the NLI evaluator on the model response would be evaluating **grounding** (a.k.a., attribution) rather than correctness. While attribution is an important topic in RAG research, we believe this is somewhat orthogonal to the goals of our paper. For example, many papers propose methods to encourage the LLM response to be grounded in the context, and hence, such methods would have a very high score for your NLI evaluator. The survey [1] has a nice overview of papers in Figure 2, and it also explicitly calls out the distinction between attribution and factuality. Automatically evaluating attribution also has many nuances, beyond just using an off-the-shelf model, as shown in [2].
> >
> > [1] [A Survey of Large Language Models Attribution](https://arxiv.org/abs/2311.03731v2), Li et al, 2023.
> >
> > [2] [AttributionBench: How Hard is Automatic Attribution Evaluation?](https://arxiv.org/abs/2402.15089), Li et al, ACL 2024.
> >
> > > Reliability of the LLM QA evaluator
> >
> > We argue that this is standard and that other papers have validated this approach in our other comment. We also show that with a deterministic metric, we can draw the same conclusions as with the LLM QA evaluator (see above).

---

> > ### Author Response · Authors · 2024-12-02
> >
> > Since this is the last day for author-reviewer discussion, we wanted to make sure you saw our comments above. We hope our additional experiments and qualitative analysis addresses your concerns.
> >
> > Specifically, we added (see [here](https://openreview.net/forum?id=Jjr2Odj8DJ&noteId=6BnELoX4N0) and [here](https://openreview.net/forum?id=Jjr2Odj8DJ&noteId=3zSSaIGhGy)):
> > 1. Clarifications about how the change in correct % was due to the different dataset, and it is not an issue with our evaluation set-up
> > 2. Experiments with a new deterministic metric ("Contains Answer") where these results verify that our main conclusions hold even if we remove the LLM QA Evaluator from the eval loop, showcasing robustness in our findings
> > 3. Several examples showing the LLM QA Evaluator is better at handling certain model responses, where it labels the response better than the deterministic method, indicating that our evaluation pipeline is intentional and rigorous
> >
> > We hope these additions ease your concerns about "how can we trust the main results of the paper?" by showing that indeed the results are trustworthy, reproducible, and robust to changes in the evaluation method!
> >
> > We are happy to answer any last questions you may have as well.

---

### Author Response · Authors · 2024-11-19

## Summary
We thank all the reviewers for their time and feedback!

We were excited to see that reviewers described our work as “address[ing] a fundamental gap in RAG literature” (BUto), “genuinely insightful and offers a number of non-trivial findings” (HFFU), and “easy to follow” (mPDo). We appreciate that they note “appropriate evaluations are used at each stage” (9Kme) and that it “presents a solid empirical foundation” (BUto).

Based on the reviewers’ helpful feedback, we have:

1. Included new experiments demonstrating the sufficient vs insufficient context insights on the human-labeled sufficient context dataset to validate findings from the autorater labels (see below)
1. Clarified the definition of sufficient context, addressed misunderstandings about our manuscript, and answered questions about our autorater (see reviewer responses).
1. Added instructions for how to try out the autorater (see below).

We look forward to answering any other questions you may have during the discussion period.

## Model Responses and Human-Labeled Sufficient Context

We provide two new tables comparing LLM responses based on human-annotated sufficient context labels. This aims to address reviewer concerns about findings based on the autorater labels. Here we can see that the results are very similar to the ones in the paper. Importantly, we draw the same conclusions when analyzing data using sufficient context labels from either our autorater or human annotators.

**Table 1: RAG Performance with Sufficient Context (54.8% of Dataset)**
| Model | %Correct | %Abstain | %Hallucinate |
|-------|-----------|-----------|--------------|
| Gemini 1.5 Pro | 84.1 | 1.6 | 14.3 |
| GPT 4o | 82.5 | 4.8 | 12.7 |
| Claude 3.5 Sonnet | 85.7 | 11.1 | 3.2 |
| Gemini 1.5 Flash | 77.8 | 4.8 | 17.5 |
| Gemma 27B | 71.4 | 3.2 | 25.4 |

Each of the models performs well on this dataset, with larger models (Gemini 1.5 Pro, GPT 4o, Claude 3.5 Sonnet) outperforming smaller models (Gemini 1.5 Flash, Gemma 27B). However, none of the models are able to perfectly answer the questions, even with sufficient context. This is consistent with our findings across the other datasets (HotpotQA, FreshQA, Musique).

**Table 2: RAG Performance with Insufficient Context (45.2% of Dataset)**
| Model | %Correct | %Abstain | %Hallucinate |
|-------|-----------|-----------|--------------|
| Gemini 1.5 Pro | 9.6 | 50.0 | 40.4 |
| GPT 4o | 23.1 | 61.5 | 15.4 |
| Claude 3.5 Sonnet | 9.6 | 53.8 | 36.5 |
| Gemini 1.5 Flash | 7.7 | 73.1 | 19.2 |
| Gemma 27B | 9.6 | 55.8 | 34.6 |

Each model is still able to answer some questions correctly given insufficient context, but also hallucinates rather than abstains across a considerable number of examples, with Hallucinations ranging from 15.4% (GPT 4o) to 40.4% (Gemini 1.5 Pro). Comparing Table 2 with Table 1, we see that models are more likely to hallucinate with insufficient rather than sufficient context.


## Try out the autorater
We also encourage reviewers to test out our autorater to get a feel for how it handles cases like multi-hop questions and out-of-domain examples.

Go to https://aistudio.google.com. For simplicity, you can use the following shorter prompt (either in the chat or in the “System Instructions”). You just need to insert the QUESTION and REFERENCES:

> You are an expert LLM evaluator that excels at evaluating a QUESTION and REFERENCES. Consider the following criteria: Sufficient Context: 1 IF the CONTEXT is sufficient to infer the answer to the question and 0 IF the CONTEXT cannot be used to infer the answer to the question. First, output a list of step-by-step questions that would be used to arrive at a label for the criteria. Make sure to include questions about assumptions implicit in the QUESTION. Include questions about any mathematical calculations or arithmetic that would be required. Next, answer each of the questions. Make sure to work step by step through any required mathematical calculations or arithmetic. Finally, use these answers to evaluate the criteria. Output the EXPLANATION (Text). Then, use the EXPLANATION to output the EVALUATION
>
> QUESTION [insert question here]
>
> REFERENCES [insert references as text here]

One very useful part of the output is the explanation, where the LLM gives a list of step-by-step reasons for how it came to the sufficient context label. We believe these reasons can also add to the utility and general applicability of a sufficient context autorater.

For RAG examples, you can copy them directly from the [Musique](https://huggingface.co/datasets/voidful/MuSiQue) dataset or from [FreshQA](https://github.com/freshllms/freshqa) without needing to download anything.

The autorater does make errors occasionally, and we do not claim that it will agree with user judgment on every example.

---

> ### Comment · Reviewer_9Kme · 2024-11-24
> **Clarification about the new results (Table 1 and 2)**
>
> Am I correct in comparing Table 1 and 2 to Figure 3 top and bottom row? If so, then I feel that the new results raise more questions about the reliability of the auto-raters.
>
> The (your) overall pipeline for categorizing questions depends on (1) sufficient context judgment and (2) LLM QA evaluator (which compares the model response against a ground truth). Errors in the two stages can cascade. Your new results mitigate (to some extent) the errors in stage 1 by using the human-rater sufficiency labels. However, they are drastically different from the results with the autorater.
>
> E.g., the rate of correct answers with insufficient labels for Gemini 1.5 Pro is 40-50%, according to Figure 3, but ~10% according to Table 2. Similar discrepancies exist in Table 1.
>
> In light of the new results, how can we trust the main results of the paper?

---

> > ### Author Response · Authors · 2024-11-28
> >
> > Thank you for your continued discussion of our paper. We agree with you that building trust in our results and resolving any discrepancies is very important. Overall, we believe that your updated concerns may be due to a misunderstanding in the results of Tables 1 and 2 in our rebuttal (which are based on a **different dataset** than the ones in Figure 3).
> >
> > We clarify these issues below, as well as reiterate the steps we have taken to verify our findings and experimental methodology. In a follow-up comment, we also demonstrate that our main findings hold when using a deterministic metric for QA Evaluation (“Contains ground truth answer”) and include references to work studying automatic evaluation metrics, hopefully alleviating your concerns about the LLM QA Evaluator.
> >
> > > Am I correct in comparing Table 1 and 2 to Figure 3 top and bottom row? . . .  However, they are drastically different from the results with the autorater. E.g., the rate of correct answers with insufficient labels for Gemini 1.5 Pro is 40-50%, according to Figure 3, but ~10% according to Table 2. Similar discrepancies exist in Table 1.
> >
> > There is actually a simple explanation for the change in correct answer rate by models compared to Figure 3: the dataset used for Tables 1 and 2 in (in the rebuttal) has a different set of questions, where the models are less much less likely to guess the right answer or have it in their parametric knowledge.
> >
> > The impact of question type is a key motivation for why we included the "correct answers with insufficient context" analysis in our submission. The rate of correct answers for insufficient context comes from a variety of sources, such as the model “guessing” the correct answer for multiple-choice or true-false questions. The examples for Tables 1 and 2 actually do not have any such questions, as they focus more on people’s names (e.g., author of X, director of Y, father of Z, etc) or other similar questions (genres, movie names, etc). This explains why the models have a lower rate of correct answers with insufficient context.
> >
> > > In light of the new results, how can we trust the main results of the paper?
> >
> > We reiterate that the difference between Tables 1 and 2 in the rebuttal vs. Figure 3 in the paper should be attributed to the fact that different datasets vary in how challenging they are for LLMs without sufficient context.
> >
> > LLM-based comparison of model response and expect response is the standard approach in many RAG evaluation systems, e.g., [RAGAS Factual Correctness](https://docs.ragas.io/en/latest/concepts/metrics/available_metrics/factual_correctness/), [DeepEval G-Eval](https://docs.confident-ai.com/docs/metrics-llm-evals), [TruLens Ground Truth Eval](https://www.trulens.org/getting_started/quickstarts/groundtruth_evals/).
> >
> > The FRAMES benchmark paper [1] also uses Gemini 1.5 Pro for LLM QA eval, and they find that “This auto-rating mechanism was tested
> > against human evaluations, in which the LLM-based evaluation showed strong alignment with human annotations (accuracy: 0.96 and Cohen’s Kappa: 0.889 for Gemini-Pro-1.5-0514 as autorating LLM), making LLM-based evaluation a suitable approach to evaluate the correctness of model responses.”
> >
> > [1] [Fact, Fetch, and Reason: A Unified Evaluation of Retrieval-Augmented Generation](https://arxiv.org/abs/2409.12941), Krishna et al, 2024.
> >
> > Hence, we believe that our use of LLM QA Evaluation is both consistent with prior work and justified based on human evaluations. In the other thread, we also show results using a deterministic metric, which leads to the same conclusions as using LLM QA Evaluation.
> >
> > While a deterministic method leads to the same conclusions, using automated rating methods was critical to our study in several ways:
> > - The LLM QA evaluator makes it more systematic to compare across LLMs because some models can be verbose or use different language conventions (see examples in the follow-up comment below). Hence, we believe more errors would be introduced if we developed ad-hoc response parsers for each model.
> > - The LLM QA evaluator can classify responses into correct, abstain, and incorrect. This is a small but important improvement in analyzing RAG systems, as deterministic methods just use correct vs. incorrect. We require abstention metrics for our application to selective generation, where we show that using the autorater’s sufficient context signal can improve selective accuracy.
> > - We can corroborate our findings across multiple datasets and models. An alternative approach would have been to use human annotators, but this is cost-prohibitive for three benchmark datasets across several LLMs.

---

> ### Author Response · Authors · 2024-11-28
>
> We also want to respond directly to your questions about LLM QA evaluation by comparing models using a common lexical matching metric. Specifically, the comparison metric uses a deterministic measure that labels LLM responses based on whether they contain the ground truth answer or not. We use LLMEval to refer to the LLM QA Evaluator, and we record the fraction of instances labeled as “Correct” by the LLM. We refer to the deterministic measure as “Contains Answer” or just “Contains” for short, and we measure the fraction of responses that contain one of the ground truth answers for the question or false otherwise.
>
> We believe that the LLM QA Evaluator should be more aligned with human expectations for labeling correctness as this task is quite simple: the model only needs to compare the LLM response to a set of possible ground truth answers. To verify this, we also include several examples below where the two metrics disagree. These examples illustrate the inherent limitations of a lexical matching approach.
>
> Here is a table of results comparing model performance on 3 datasets, split based on our sufficient context autorater. This directly matches the setup for the results in Figure 3, where the LLMEval numbers correspond to the correct percentage (green bars in Figure 3).
>
> | Model | Context | FreshQA |  | Musique |  | HotpotQA |  |
> | --- | --- | --- | --- | --- | --- | --- | --- |
> | | | Contains | LLMEval | Contains | LLMEval | Contains | LLMEval |
> | --- | --- | --- | --- | --- | --- | --- | --- |
> | Gemini 1.5 Pro | Suff | 80.3% | 89.1% | 60.1% | 83.4% | 47.6% | 67.5% |
> |  | Insuff | 31.4% | 41.2% | 33.6% | 49.5% | 34.2% | 49.4% |
> | GPT 4o | Suff | 84.3% | 89.1% | 64.6% | 83.4% | 52.4% | 71.9% |
> |  | Insuff | 36.3% | 44.1% | 44.4% | 61.4% | 46.1% | 59.5% |
> | Gemma 27B | Suff | 26.9% | 27.5% | 10.8% | 23.3% | 40.7% | 64.1% |
> |  | Insuff | 11.8% | 6.9% | 7.2% | 10.1% | 22.7% | 37.9% |
> | Claude 3.5 Sonnet | Suff | 67.9% | 73.1% | 48.9% | 74.0% | 46.3% | 66.7% |
> |  | Insuff | 26.5% | 33.3% | 19.9% | 40.4% | 29.0% | 38.3% |
>
>
> The ‘Contains Answer’ evaluation is generally stricter than the LLM Eval, as it misses correct answers with different formatting. For example:
>
> ```
> Question: What date did the creator of Autumn Leaves die?
> Ground Truth Answer: 13 August 1896
> Model Response: August 13, 1896.
> Contains Answer: False
> LLM Eval: Correct
> ```
>
> Checking ‘Contains Answer’ also fails on answers that don’t match exactly but are semantically correct. For example:
>
> ```
> Question: What former Los Angeles Lakers majority owner is the father of Jeanie Marie Buss?
> Ground Truth Answer: Gerald Hatten Buss
> Model Response: Jerry Buss.
> Contains Answer: False
> LLM Eval: Correct
> ```
> Using ‘Contains Answer’ also occasionally marks answers as “Correct” due to the nature of the response. This happens more frequently with smaller models (e.g., Gemma), which are more likely to output poorly formed answers. This example is from Gemma 27B (insufficient context)
>
> ```
> Question: What is Amazon Prime Video's most watched premiere ever?
> Ground Truth: The Rings of Power
> Model Response: The series explores the forging of the Rings of Power, the rise of Sauron, and the last alliance of Elves and Men.
> Contains Answer: True
> LLM Eval: Hallucinate
> ```
>
> Two of our main insights are that (i) LLMs hallucinate quite a bit even with sufficient context, and (ii) LLMs struggle to abstain with insufficient context, which leads to a much higher hallucination rate with insufficient context. Importantly, our conclusions about both (i) and (ii) remain true even if we account for rater errors. Moreover, if we use `Contains Answer’ as the response evaluator, then we still see (i) and (ii). Together, these build trust in our paper’s conclusions.

---

### Meta-Review · Area_Chair_Kdix · 2024-12-21

**Metareview:**

Summary of the paper: This paper investigates the behavior of LLMs in RAG systems, focusing on the concept of "sufficient context"—defined as context that unambiguously supports an answer to a given question. The authors develop an "autorater" prompt to evaluate context sufficiency, validating it on a set of 115 hand-labeled (question, context, answer) triplets. The study includes a detailed analysis of various QA datasets. The authors explore the implications of context sufficiency on the performance of different LLMs in RAG systems, noting that closed-source models (like Gemini and GPT) often fail to abstain when the context is lacking, leading to incorrect answers. In contrast, open-source models are more prone to hallucination or abstaining when context is adequate. To address these issues, the authors propose a selective generation technique that utilizes context sufficiency signals to improve response accuracy, achieving enhancements of up to 10% in models like Gemini and GPT. The findings emphasize the importance of context sufficiency in optimizing LLM performance and reducing the incidence of hallucinations in high-stakes applications.

Strengths of the paper:
- Innovative Concept of "Sufficient Context": The formalization of sufficient context addresses a gap in RAG literature, improving the evaluation of LLM performance.
- Rigorous Method and Evaluation: The creation and testing of a sufficient context "autorater" across various LLMs, along with extensive evaluation metrics, establish a solid empirical foundation and demonstrate strong generalizability.
- Practical Framework for Hallucination Mitigation: The selective generation framework effectively combines context sufficiency with model confidence, offering a practical approach to reduce hallucinations in LLMs.

Weaknesses of the paper:
- Clarity of Presentation (Reviewer mPDo, 9Kme): The presentation of the paper requires improvement, particularly regarding the core concept of "sufficient context," which needs clarification. Additionally, there are minor writing issues may obstruct reproducibility and imprecise formulations that should be addressed.
- Lack of Discussion on QA Benchmarks (Reviewer 9Kme): The paper should provide a clear discussion of the various QA benchmarks and their implications. This would help explain why different findings are not directly comparable and highlight the nuances of how certain QA datasets behave differently.
- Concerns About Automated Metric Reliability (Reviewer 9Kme): The paper should include a quantitative analysis of the ways in which QA models perform correctly, even with insufficient context. This would enhance the discussion on the reliability of automated metrics, moving beyond anecdotal evidence.
- Inadequate Explanation of Autorater (Reviewer BUto, HFFU): The design and reliability of the autorater, especially in multi-hop contexts, are not thoroughly explained. The paper would benefit from including qualitative examples to illustrate its effectiveness.
- Dataset Accessibility (Reviewer HFFU): The dataset used for analysis is not publicly available, which limits the reproducibility of the study.
- Other Minor Issues (Reviewer BUto): There is a limited discussion on computational overhead, a lack of insight into the impact of retrieval strategies, and a need for more detailed insights specific to the models used.

Reasons for decision: I believe that the authors have adequately addressed most of the concerns raised by the reviewers during the rebuttal. The majority of reviewers agree that this paper should be accepted at ICLR. After carefully considering all the comments from the author-reviewer discussion and discussing with reviewer pMDo, it is noted that the contribution of the paper is seen as incremental. This perception stems partly from the reviewers' view that the analysis presented in the paper is not particularly exciting, which may be attributed to its presentation. However, I personally align with other reviewers who recognize that the analysis surrounding the core concept of "sufficient context" offers valuable insights that are currently lacking in existing RAG research. This work has the potential to engage the broader community and make a significant impact (Of course this paper needs polish based on the reviewer-author discussions). Therefore, I am inclined to recommend the acceptance of this paper.

**Additional Comments On Reviewer Discussion:**

I think the authors have adequately addressed most of the concerns raised by the reviewers during the rebuttal:
- Clarified the definition of sufficient context, addressed misunderstandings about the paper, and answered questions about our autorater.
- Included new experiments demonstrating the sufficient vs insufficient context insights on the human-labeled sufficient context dataset to validate findings from the autorater labels.

---

### Decision · Program_Chairs · 2025-01-22

Accept (Poster)